# *Pf*MORC protein regulates chromatin accessibility and transcriptional repression in the human malaria parasite, *Plasmodium falciparum*

Zeinab M Chahine[1†], Mohit Gupta[1†‡], Todd Lenz[1†], Thomas Hollin[1†], Steven Abel[1], Charles Banks[2], Anita Saraf[2§], Jacques Prudhomme[1], Suhani Bhanvadia[1], Laurence A Florens[2], Karine G Le Roch[1*]

[1]Department of Molecular, Cell and Systems Biology, University of California, Riverside, Riverside, United States; [2]Stowers Institute for Medical Research, Kansas City, United States

*For correspondence:
karine.leroch@ucr.edu

[†]These authors contributed equally to this work

Present address: [‡]Cancer Early Detection Advanced Research Center, Oregon Health and Science University, Portland, United States; [§]Shankel Structural Biology Center, The University of Kansas, Lawrence, United States

Competing interest: The authors declare that no competing interests exist.

## eLife Assessment

This **important** study examines the role of Microrchidia (MORC) proteins in the human malaria parasite *Plasmodium falciparum*. **Solid** experimental results, including genome editing and chromatin profiling methods (ChIP-seq and Hi-C), provide a comprehensive picture of the critical role MORC plays in shaping parasite chromatin. Depletion of MORC results in a lethal collapse of heterochromatin and parasite death, nominating the factor as a new target of antimalarial therapies.

## Abstract

The environmental challenges the human malaria parasite, *Plasmodium falciparum*, faces during its progression into its various lifecycle stages warrant the use of effective and highly regulated access to chromatin for transcriptional regulation. Microrchidia (MORC) proteins have been implicated in DNA compaction and gene silencing across plant and animal kingdoms. Accumulating evidence has shed light on the role MORC protein plays as a transcriptional switch in apicomplexan parasites. In this study, using the CRISPR/Cas9 genome editing tool along with complementary molecular and genomics approaches, we demonstrate that *Pf*MORC not only modulates chromatin structure and heterochromatin formation throughout the parasite erythrocytic cycle, but is also essential to the parasite survival. Chromatin immunoprecipitation followed by deep sequencing (ChIP-seq) experiments suggests that *Pf*MORC binds to not only sub-telomeric regions and genes involved in antigenic variation but may also play a role in modulating stage transition. Protein knockdown experiments followed by chromatin conformation capture (Hi-C) studies indicate that downregulation of *Pf*MORC impairs key histone marks and induces the collapse of the parasite heterochromatin structure leading to its death. All together these findings confirm that *Pf*MORC plays a crucial role in chromatin structure and gene regulation, validating this factor as a strong candidate for novel antimalarial strategies.

## Introduction

Malaria is a mosquito-borne infectious disease that is caused by protozoan parasites of the genus *Plasmodium*. Among the five human-infecting species, *Plasmodium falciparum* is the deadliest, with over 619,000 deaths in 2023 (**WHO, 2023**). To adapt to extreme environmental challenges, *P. falciparum* possesses unique strategies that direct the tightly coordinated changes in gene

**eLife digest** Malaria is an infectious disease caused by parasites that spread to humans through mosquitoes. The parasite species *Plasmodium falciparum* accounts for the most fatal forms of the disease and can undergo substantial genetic changes that allow it evade detection by the human immune system. A better understanding of how *P. falciparum* controls its gene expression could help find new ways for treating this aggressive form of malaria.

One of the key mechanisms by which cells – including parasites – control gene expression is by remodeling the structure of their DNA, which is tightly packed in to a construct known as chromatin. By loosening or condensing regions of chromatin, cells can make certain genes more or less accessible to the machinery responsible for activating them.

Recently, a protein called MORC has been found to play a key role in chromatin remodeling and gene expression in different parasites, plants and animals. However, it remains unclear if MORC also influences chromatin structure of *P. falciparum* and contributes to the parasite's ability to evade immune detection.

To investigate this question, Chahine, Gupta, Lenz, Hollin et al. employed a variety of gene editing tools, including CRISPR/Cas9. This revealed that reducing MORC levels caused the structure of chromatin in *P. falciparum* to collapse, ultimately resulting in the death of the parasite. The team also found that MORC not only directly interacts with the parasite's chromatin, but also other factors involved in chromatin remodeling and genes that help the parasite evade the immune system.

These findings highlight the essential role of MORC protein in maintaining chromatin structure and the survival of *P. falciparum*, and offer a new therapeutic target for controlling the spread of this aggressive form of malaria.

expression and control transcriptional switching in genes encoded by multigene families to ensure antigenic variation and immune evasion. Changes in gene expression throughout the parasite life cycle are controlled by a surprisingly low repertoire of transcription factors (TFs) encoded in the *Plasmodium* genome (*Coulson et al., 2004*; *Balaji et al., 2005*; *De Silva et al., 2008*; *Yuda et al., 2009*; *Iwanaga et al., 2012*; *Sinha et al., 2014*; *Kafsack et al., 2014*; *Lesage et al., 2018*). The 27 apicomplexan APETALA2 (ApiAP2) DNA-binding proteins are the only well-documented parasite TFs known to contribute to the modulation of gene expression throughout various stages of the parasite's development (*Balaji et al., 2005*; *Yuda et al., 2009*; *Iwanaga et al., 2012*; *Sinha et al., 2014*; *Kafsack et al., 2014*; *Yuda et al., 2010*; *Modrzynska et al., 2017*; *Gu et al., 2017*; *Yuda et al., 2020*). Since the discovery of the AP2 gene family (*Balaji et al., 2005*), evidence has alluded to their role as master regulators of gene expression throughout transitory phases of the parasite life cycle. The most well-documented are the gametocyte-specific TFs (AP2-G, AP2-G2) (*Sinha et al., 2014*; *Yuda et al., 2015*) with the subsequent discovery of those associated with a transition to sexual differentiation (*Yuda et al., 2020*) as well as sporozoite (AP2-SP) and liver stages (AP2-L) (*Iwanaga et al., 2012*; *Yuda et al., 2010*). The *Plasmodium* genome, however, encompasses well over 5,000 protein-coding genes suggesting there are most likely other molecular components responsible for gene expression. It is believed that, to offset the relatively low TF range, *Plasmodium* has evolved additional mechanisms regulating gene expression including mechanisms that use epigenetics factors, RNA binding proteins, or regulate chromatin structure (*Bozdech et al., 2003b*; *Le Roch et al., 2003*; *Bozdech et al., 2003a*; *Srinivasan et al., 2004*; *Silvestrini et al., 2005*; *Yang et al., 2021*). All together these components work in combination to regulate the dynamic organization of DNA, and the cascade of gene expression required for the parasite life cycle progression (*Dekker et al., 2002*; *Dostie et al., 2006*; *Cui et al., 2008*; *Dixon et al., 2012*; *Dembélé et al., 2014*; *Deng et al., 2015*; *Bunnik et al., 2019*; *Batugedara et al., 2020*). Despite their importance, the identification and functional characterization of regulatory players controlling chromatin remains a challenge in this intractable organism. However, with the advent of sensitive technologies capable of capturing important chromatin associated regulatory complexes such as ChIP, chromatin conformation capture (Hi-C) technologies, and chromatin enrichment for proteomics (ChEP), we can now solve important facets of molecular components controlling epigenetics and chromatin structure.

MORC belongs to a highly conserved nuclear protein superfamily with widespread domain architectures that link MORCs with signaling-dependent chromatin remodeling and epigenetic regulation across plant and animal kingdoms, including the apicomplexan parasites (*Lorković, 2012*; *Li et al., 2013*; *Weiser et al., 2017*; *Singh et al., 2021a*). In all organisms, MORC contains several domains that form a catalytically active ATPase. Apicomplexan MORC ATPases are encircled by Kelch-repeat β-propellers as well as a CW-type zinc finger domain functioning as a histone reader (*Farhat et al., 2020*). In higher eukaryotes, MORCs were first identified as epigenetic regulators and chromatin remodelers in germ cell development. Currently, these proteins are shown to be involved in various human diseases including cancers and are expected to serve as important biomarkers for diagnosis and treatment (*Wang et al., 2021*). In the apicomplexan parasite, *Toxoplasma gondii*, *Tg*MORC was shown to recruit the histone deacetylase HDAC3 to particular genome loci to regulate chromatin accessibility, restricting sexual commitment (*Farhat et al., 2020*; *Zhong et al., 2023*). *Tg*MORC-depleted cells also resulted in change in a gene expression with up regulation of secondary AP2 factors and a significant shift from asexual to sexual differentiation (*Singh et al., 2021a*; *Farhat et al., 2020*; *Saksouk et al., 2005*; *Bougdour et al., 2009*; *Hillier et al., 2019*). Having multiple homologs with *T. gondii*, *Plasmodium* AP2 protein conservation is primarily restricted to their AP2 DNA-binding domains (*Jeninga et al., 2019*; *Hiyoshi and Wada, 1990*). Recently, ChEP experiments done in *P. falciparum* identified *Pf*MORC as one of the highest enriched chromatin-bound proteins at different stages of the parasite intraerythrocytic development cycle (IDC) (*Batugedara et al., 2020*). *Pf*MORC was also detected at a relatively high level throughout the parasite life cycle, including sporozoites and liver stages (*Farhat et al., 2020*; *Singh et al., 2021b*), and has been identified in several protein pull-down experiments as targeting both AP2 TFs and epigenetic factors (*Singh et al., 2021a*; *Farhat et al., 2020*; *Hillier et al., 2019*; *Subudhi et al., 2023*; *Bryant et al., 2020*; *Singh et al., 2024*). Immunoprecipitation experiments demonstrated that *Pf*MORC seems to interact with AP2-G2 (*Singh et al., 2021a*), a TF that plays a critical role in the maturation of gametocytes. However, the genome-wide distribution and exact function of this protein throughout the parasite life cycle remained elusive.

In this study, we apply CRISPR/Cas9 genomic editing technologies to determine the function, genome distribution, and indispensability of *Pf*MORC throughout the parasite's IDC. Immunoprecipitation of an HA-tagged parasite line validates the role of *Pf*MORC in heterochromatin structure maintenance. Using the downregulation of *Pf*MORC induced through the TetR-DOZI system, we demonstrate the functional significance of *Pf*MORC in heterochromatin stability and gene repression. Immunofluorescence-based assays and ChIP-seq experiments show that *Pf*MORC localizes to heterochromatin clusters at or near *var* genes with significant overlap with well-known signatures of the parasite heterochromatin post-translational modification (PTM) H3K9-trimethylation (H3K9me3) marks. When *Pf*MORC was downregulated, a level of H3K9me3 was detected at a lower level, demonstrating a possible role of this protein in epigenetic regulation and gene repression. Finally, Hi-C analyses demonstrate that downregulation of *Pf*MORC results in significant dysregulation of chromatin architectural stability resulting in parasite death. All together our work provides significant insight into the role of *Pf*MORC in the maintenance of the pathogens' epigenetics and chromosomal architectural integrity and validates this protein as a promising target for novel therapeutic interventions.

## Results

### Generation of *Pf*MORC-HA transgenic line shows nuclear localization in heterochromatin clusters

To characterize the role of *Pf*MORC in *P. falciparum*, we applied the CRISPR/Cas9 genome editing tool to add a 3 X-HA tag at the C-terminal coding region of the *Pfmorc* locus (PF3D7_1468100) in an NF54 line (*Ganesan et al., 2016*, *Figure 1a*, *Supplementary file 1*). Recovered transgenic parasites were cloned using limiting dilution and the correct incorporation of the tag into the parasite genome was validated via PCR and whole genome sequencing (WGS) (*Figure 1b*, *Supplementary file 2*). WGS analysis confirmed the presence of the HA tag at the expected locus with no obvious off-target effect but uncovered an additional nonsense mutation in the *gametocyte development protein 1* (*gdv1*) gene (PF3D7_0935400) at amino acid position 561 out of 599 total. Mutations in *gdv1* have been detected in the past and seem to emerge relatively often, indicating a fitness benefit in the in vitro culture system used in the laboratory (*Kafsack et al., 2014*). The involvement of GDV1 as essential

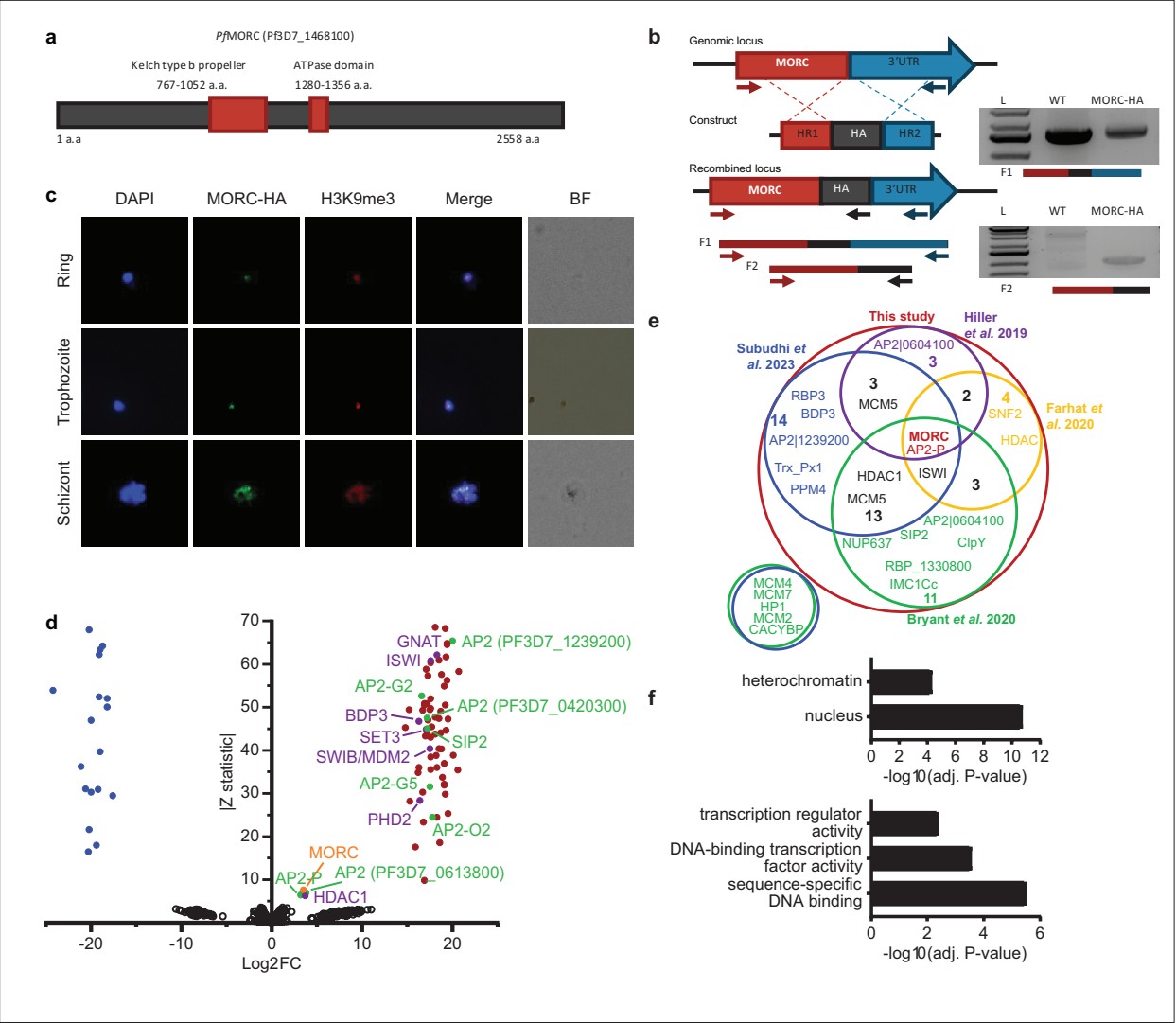

**Figure 1.** *Pf*MORC-HA is associated with heterochromatin. (**a**) Illustration of the *Pf*MORC containing domains including Kelch type b propeller and ATPase domains using InterProScan. (**b**) Design strategy applied for *Pf*MORC C-terminal HA tagging. PCR amplification of the genomic C-terminus end of *Pfmorc* region extending towards the 3'UTR (**F1**) as well as extension from the C-terminus towards the HA flanking sequence (**F2**) verifies the correct insertion site. NF54 genomic DNA was used as a negative control. (**c**) Immunofluorescence analysis (IFA) experiment: *Pf*MORC foci (green) expressing co-localization with H3K9me3 marks (red). Cell nuclei are stained with DAPI (blue). BF: brightfield (**d**) Protein immunoprecipitation: Significance plot representing *Pf*MORC interactome recovered through immunoprecipitation followed by mass spectrometry (IP-MS). Graph lists Microchidia (MORC) (orange) bindings partners of the highest affinity associated with TF regulation (green) and chromatin remodelers, erasers, and writers (purple). Proteins enriched in the *Pf*MORC-HA samples compared with controls were filtered with log$_2$ FC ≥2 and Z statistic >5. (**e**) Venn diagram representing overlapping proteins identified among five publications. Values represent the total number of significant proteins identified as overlapping between two subsets. (**f**) Gene ontology enrichment analysis of the significantly enriched proteins. The top two terms of Cellular Component (top) and top three terms of molecular function (bottom) are represented as -Log$_{10}$ (adjusted P-value) (Fisher's exact test with Bonferroni adjustment). Figure 1b was created with BioRender.com.

The online version of this article includes the following source data and figure supplement(s) for figure 1:

**Source data 1.** Original, unedited gel image for displayed *Pf*MORC-HA results displayed in *Figure 1*.

**Source data 2.** Original gel image for displayed *Pf*MORC-HA results displayed in *Figure 1* with marked ladder and described lanes.

**Figure supplement 1.** Expression of Microchidia (MORC) in *pDC2*-MORC-HA lines.

in sexual commitment is well substantiated and further study involving the role of *Pf*MORC in sexual commitment could not be fully addressed (*Sinha et al., 2014*; *Kafsack et al., 2014*; *Yuda et al., 2015*; *Filarsky et al., 2018*; *Yuda et al., 2021*; *Kent et al., 2018*, *Supplementary file 2*). Despite this finding, we were able to design experiments to determine the role of *Pf*MORC throughout the IDC

as no other major detrimental mutations were detected in the genome. We first validated *Pf*MORC protein expression through Western blot analysis. Regardless of potential protein degradation, results showed an expected band size within the ~290 kDa range that was absent in NF54 control compared to our tagged line (*Figure 1—figure supplement 1*). Immunofluorescence (IFA) analysis of intracellular parasites revealed that *Pf*MORC is localized in the nucleus throughout the parasite IDC in punctate patterns (*Figure 1c*). A single foci per parasite was detected in the nuclear periphery at the ring stage. An increased number of foci with a more diffuse signal could be detected as the number of DNA copies increased at the schizont stage. Moreover, punctuates were shown to have strong colocalization signals with histone H3K9me3 marks present at the different parasite developmental stages analyzed. This was most evident during the early ring stage of the parasite IDC where the chromatin organization is more compact with a single DNA copy. Altogether, our findings indicated that *Pf*MORC is a true nuclear protein that is most likely associated with *P. falciparum* heterochromatin cluster(s).

## *Pf*MORC interacting partners include proteins involved in heterochromatin maintenance, transcription regulation, and chromatin remodeling

To confirm the association of *Pf*MORC with proteins involved in heterochromatin cluster(s), we performed immunoprecipitation (IP) followed by mass-spectrometry (MS) analysis using mature stages of *Pf*MORC-HA and parental line as control. MORC was detected with the highest peptides (97 and 113) and spectra (1041 and 1177) counts compared to control conditions (5 and 7 peptides; 16 and 43 spectra) confirming the efficiency of our pull-down. However, considering the relatively large size of the MORC protein (295 kDa) and its weak detection in the control, the $\log_2$ FC and Z-statistic after normalization are minimal when compared to smaller proteins that were not identified in the control samples. To define a set of *Pf*MORC-associated proteins, we used the QPROT statistical framework (*Choi et al., 2015*) to compare proteins detected in *Pf*MORC samples and controls. We identified 73 *Pf*MORC-associated proteins ($\log_2$ FC ≥2 and Z statistic >5) (*Figure 1d*, *Supplementary file 3*). Additional chromatin remodeling proteins were detected such as ISWI chromatin-remodeling complex ATPase (PF3D7_0624600) and SWIB/MDM2 domain-containing protein (PF3D7_0611400), as well as the epigenetic readers BDP3 (PF3D7_0110500) and PHD2 (PF3D7_1433400) (*Hoeijmakers et al., 2019*). We also detected a significant enrichment of histone erasers and writers such as a N-acetyltransferase (PF3D7_1020700), HDAC1 (PF3D7_0925700), and SET3 (PF3D7_0827800). Several of which have been detected to interact with MORC in previous studies including the AP2-G5, HDAC1, ELM2, and ApiAp2 proteins (*Farhat et al., 2020*; *Subudhi et al., 2023*; *Bryant et al., 2020*; *Singh et al., 2024*, *Figure 1e*). Notably, we observed that many of the detected proteins were enriched at levels comparable to those of the bait proteins such as AP2-P and HDAC1. This may be a result of proteins forming complexes in a one-to-one ratio, leading to similar enrichment levels. Notably, two of these three proteins have been reported to interact with MORC in several studies, further supporting a strong interaction between them. Gene Ontology (GO) enrichment analysis revealed that these proteins are nuclear and are associated with heterochromatin, DNA binding, and transcription factor activity (*Figure 1f*). Detailed analysis showed the detection of SIP2, involved in heterochromatin formation and chromosome end regulation (*Flueck et al., 2010*), and eight AP2 transcription factors, including AP2-G2, AP2-O2, AP2-G5. Among the other AP2s detected, five were previously identified by ChIP-seq as *Plasmodium falciparum* AP2 Heterochromatin-associated Factors (*Pf*AP2-HFs) (*Shang et al., 2022*). Our findings, corroborated with several published works, indicate that *Pf*MORC interacts with multiple ApiAP2 TFs, chromatin remodelers, and epigenetic players associated with heterochromatin regions.

## Genome-wide distribution of *Pf*MORC displays functional roles in stage-specific gene silencing and antigenic variation

Given the multifunctional role of MORC proteins throughout apicomplexan parasites, we further explored the association of *Pf*MORC with chromatin accessibility and stage-specific gene regulation by assessing the genome-wide distribution of *Pf*MORC using chromatin immunoprecipitation followed by deep sequencing (ChIP–seq) in duplicate at the ring, trophozoite, and schizont stages. Inspection of *Pf*MORC binding sites across each chromosome revealed a strong signal for telomeric and subtelomeric regions of the parasite genome, as well as the internal clusters of genes responsible

for antigenic variation such as *var* and *rifin* (**Figure 2a–c**, **Supplementary file 4**). We found that while only 47% of reads mapped to antigenic genes in the ring stage, 95% of reads mapped to the same genes during the trophozoite stage and 82% during the schizont stage. Although a vast majority of the genome is actively transcribed during the trophozoite stage, these antigenic gene families are under tight regulatory control to ensure mutually exclusive expression and participate in immune evasion (**Chen et al., 1998**; **Scherf et al., 1998**). This could explain the disparity in the level of *Pf*MORC binding of the coding regions between the ring and trophozoite stages (**Figure 2c**, **Figure 2—figure supplement 1b**).

We also evaluated *Pf*MORC binding of stage-specific gene families including gametocyte-related genes, and merozoite surface proteins (*msp*) (**Meerstein-Kessel et al., 2018**). In trophozoite and schizont stages, gametocyte-associated genes contain a mean of <0.5 RPKM normalized reads per nucleotide of *Pf*MORC binding within their promoter region, whereas antigenic gene families such as *var* and *rifin* contain ~1.5 and 0.5 normalized reads, respectively (**Figure 2b**). The difference is even greater within the gene body with gametocyte genes displaying almost no reads mapped more than 200 bp downstream of the TSS and antigenic gene families containing 0.5–1.5 RPKM normalized reads. However, the gametocyte-specific transcription factor AP2-G, known to be repressed during the IDC and required for sexual commitment, deviates from this trend and contains similar levels of *Pf*MORC binding to *var* genes (**Figure 2c** and **Figure 2—figure supplement 1b, c**). These results indicate a major role of *Pf*MORC in controlling AP2-G and sexual differentiation. For some of the *msp* genes that are usually expressed later during the IDC to prepare the parasite for egress and invasion of new erythrocytes, *Pf*MORC binding was detected in the gene bodies at the trophozoite stage. We also detected a small switch in *Pf*MORC binding sites, moving from their gene bodies to their intergenic regions at the schizont stage (**Figure 2—figure supplement 1c**). *Pf*MORC most likely moved away from the gene body to the regulatory regions surrounding the transcription start site (TSS) to guide RNA Polymerase and transcription factors and aid in activating expression of genes at the schizont stage that are crucial for egress and invasion. These results provide strong evidence for the direct effects that *Pf*MORC binding has on tightly controlled antigenic and stage-specific genes including crucial invasion and gametocyte genes.

## *Pf*MORC is essential for *P. falciparum* survival

We next sought to confirm the functional relevance of *Pf*MORC protein in parasite survival. While partial *Pf*MORC knockdown using the glmS-ribozyme system had previously been shown to have no significant effect on the parasite survival (**Singh et al., 2021b**), another study using transposon mutagenesis (*piggyBac*) identified *Pf*MORC as likely essential (**Zhang et al., 2018**). To resolve these conflicting results, we applied a complementary approach using the CRISPR/Cas9 gene editing strategy to incorporate an inducible TetR-DOZI system to knockdown (KD) *Pf*MORC (**Goldfless et al., 2014**) through the administration of anhydrotetracycline (aTC) (**Figure 3a**, **Ganesan et al., 2016**; **Nasamu et al., 2021**). The protein was also modified to include a C-terminal 3 x-HA tag. Parental and transgenic clones were validated via PCR (**Figure 3b**) and WGS to confirm the correct insertion of the inducible system (**Supplementary file 1 and 2**) as well as the absence of major mutation that could explain some of the phenotypes observed. While results from our WGS validated our editing strategy without the detection of any obvious deleterious off-target effect, we identified a nonsense mutation in the AP2-G gene (PF3D7_1222600) explaining our inability to obtain mature gametocytes. Protein expression and successful KD was confirmed via western blot with a decrease of ~58% and ~88.2% in *Pf*MORC expression at 24 hpi and 36 hpi, respectively, compared to their aTC-supplemented conditions (**Figure 3—figure supplement 1**). Downregulation of *Pf*MORC was detected significantly above the level observed by Singh and colleagues (**Singh et al., 2021b**).

We then performed a phenotypic assay on (+/-) aTC *Pf*MORC-HA-TetR-DOZI parasite cultures. The assay was conducted in replicates of synchronized cultures that were then split into (+) aTC and (-) aTC conditions either at the ring or trophozoite stages on *Pf*MORC-HA-TetR-DOZI clones and WT control lines (**Figure 3c–f**, **Supplementary file 5**). Sequential morphological screens were performed through Giemsa-stained smears and monitored by microscope imaging. As opposed to what was observed previously with the glmS-ribozyme system (**Singh et al., 2021b**), aTC removal at the ring stage induced clear signs of stress and cell cycle arrest in mid- trophozoite and schizont stages of the first intraerythrocytic cycle (**Figure 3c–d**) compared to (+) aTC *Pf*MORC and WT parasites. Our data

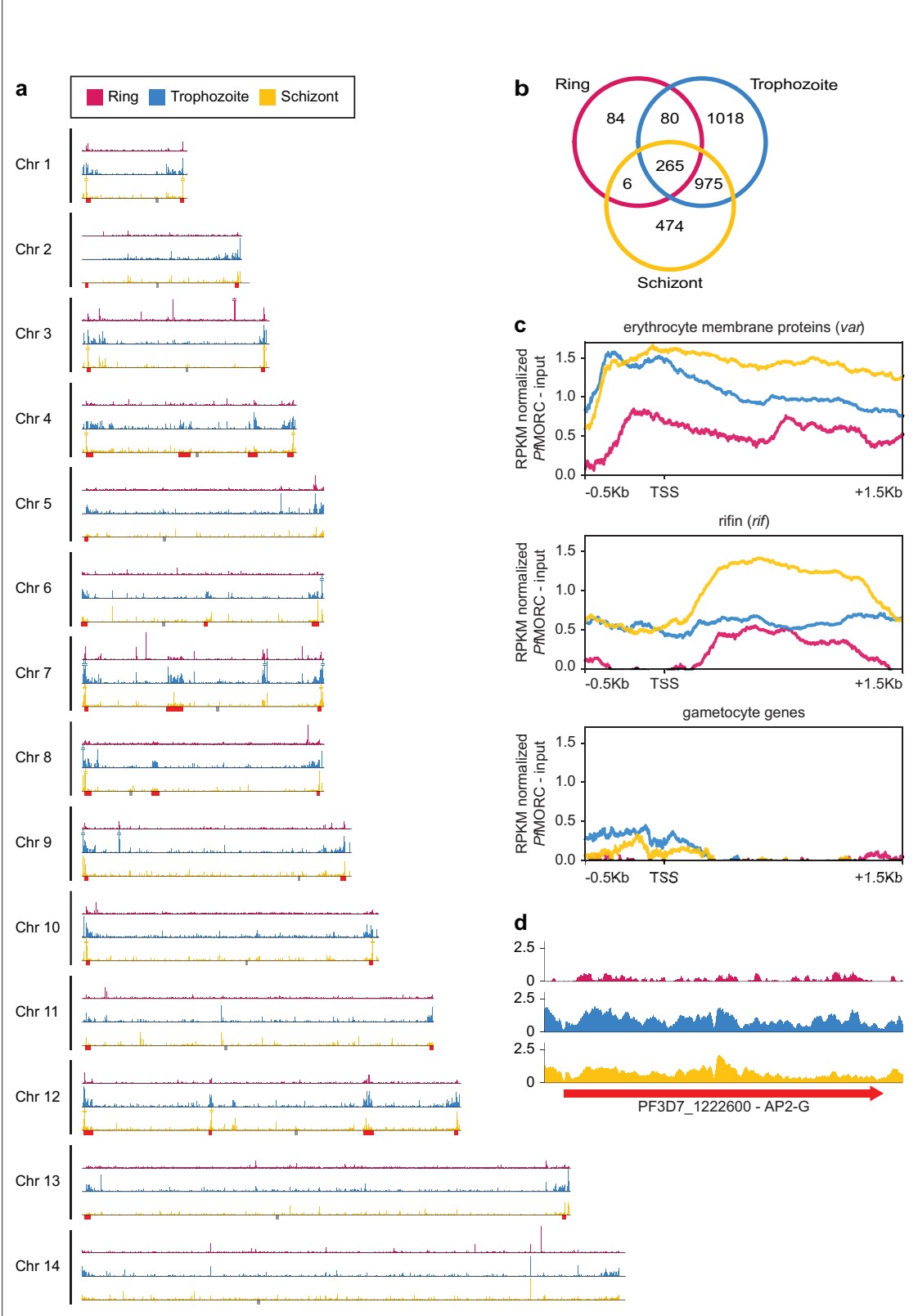

**Figure 2.** Genome-wide distribution of *Pf*MORC proteins. (**a**) Chromosome distribution plots of *Pf*MORC binding show a predisposition for subtelomeric and internal *var* gene regions (red). Each track is input subtracted, and the per-million read count is normalized before normalizing the track height to allow for direct comparison between stages. Gray boxes indicate the position of centromeres. (**b**) Overlap of called peaks between time points. (**c**) Profile plots showing *Pf*MORC coverage from 0.5 kb 5′ of the transcription start site (TSS) to 1.5 kb 3′ of the TSS in the ring, trophozoite, and

*Figure 2 continued on next page*

*Figure 2 continued*

schizont stages. Each plot includes per-million read count normalized coverage at 1 bp resolution for all genes within the *var* and *rifin* gene families as well as gametocyte-specific genes. (**d**) *Pf*MORC coverage of the gametocyte-specific transcription factor *ap2-g*.

The online version of this article includes the following figure supplement(s) for figure 2:

**Figure supplement 1.** Correlation, peak calling, and gene family-specific coverage of *Pf*MORC Chromatin immunoprecipitation followed by deep sequencing (ChIP-seq).

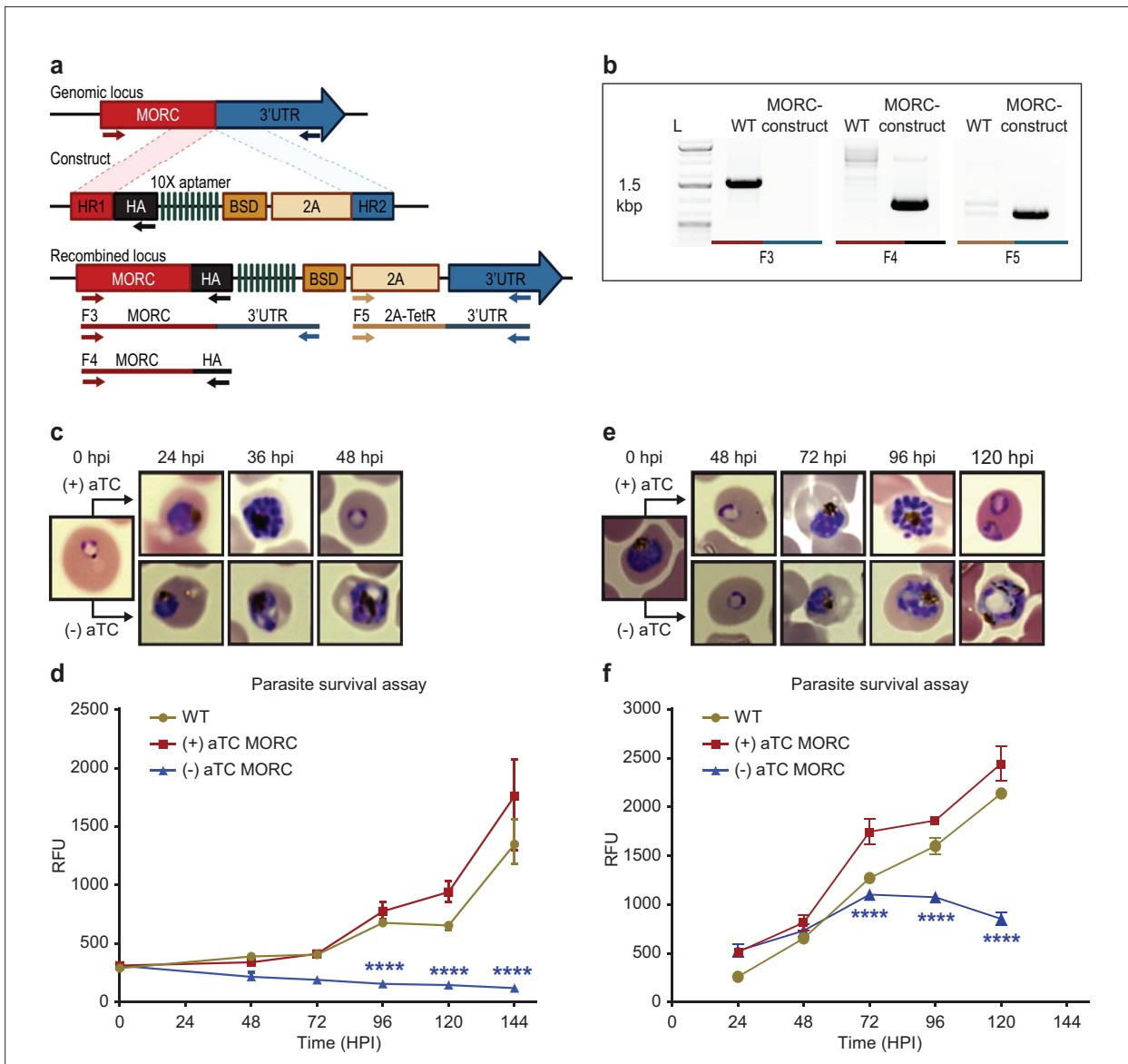

**Figure 3.** *Pf*MORC is essential for cell survival. (**a**) Diagram representation of *Pf*MORC-HA-TetR-DOZI plasmid. (**b**). PCR amplification is used to verify genomic insertion using primers sets targeting 1.5 kbp of WT *Pfmorc* genome locus absent in transgenic line (Microchidia, MORC construct) (F3) as well as verification of HA insertion (F4) and TetR-DOZI system extending along 3′ UTR of the construct (F5). (**c**) Phenotypic and (**d**) quantitative analysis of parasite cell progression after aTC withdrawal at the ring stage (0–6 hpi) (2-way ANOVA, n=3, p≤0.0001). (**e**) Phenotypic and quantitative (**f**) analysis of parasite cell progression after aTC removal at the trophozoite stage of cell cycle progression (24 hpi) (two-way ANOVA, n=3, p≤0.0001). Figure 3a was created with BioRender.com.

The online version of this article includes the following figure supplement(s) for figure 3:

**Figure supplement 1.** Expression of *Pf*MORC-HA in *Pf*MORC-HA-TetR-DOZI lines.

showed a ~53%, 77%, and 84% (n=3, p≤0.0001) drop in parasitemia at 48-, 72-, and 120- hr post-invasion (hpi), respectively, when compared to control conditions. aTC supplemented and WT cultures showed unperturbed cell cycle progression, reinvasion, and morphological development. Interestingly, when aTC was withheld at the late trophozoite stages (24–30 hpi), parasites could complete the first cycle and successfully reinvade into new red blood cells (RBCs) (*Figure 3e–f*). The evident hindered phenotypic response may either be associated with the delay time for the protein to be completely depleted or may suggest a more dispensable role of *Pf*MORC in the schizont stage. However, clear indications of stress and cell cycle arrest were ultimately detected at trophozoite and early schizont stages of the second cell cycle. Quantitative analysis of (-) aTC *Pf*MORC cultures revealed significantly decreased parasitemia of ~37% and 66% (n=3, p≤0.0001) at 96 hpi and 120 hpi, respectively, compared to (+) aTC *Pf*MORC and WT control lines (*Figure 3e–f*, *Supplementary file 5*), confirming the importance of the protein at the trophozoite and early schizont stages.

## Effect of *Pf*MORC knockdown on parasite transcriptome

To define the effects of *Pf*MORC KD on transcription, we conducted detailed time-course measurements of mRNA levels using RNA-sequencing (RNA-seq) throughout the parasite asexual cycle on (+/-) aTC *Pf*MORC lines. Parasites were first synchronized and aTC was removed from one set of samples. Total RNA was then extracted at the trophozoite (24 hpi) and the schizont (36 hpi) stages to allow for the detection of early and late changes in gene expression after aTC removal. Pairwise correlations analysis (*Figure 4—figure supplement 1*) between (+/-) aTC *Pf*MORC treated lines at the two different time points confirmed the appropriate reproducibility of our RNA-seq experiments.

Our results identified a relatively low number of differentially expressed genes at 24 hpi with 96 and 93 upregulated and downregulated genes, respectively (FDR <0.05 and $\log_2$ FC >0.5 or <–0.5) (*Figure 4a*, *Supplementary file 6*). This data is in agreement with the absence of major phenotypic changes observed in (-) aTC *Pf*MORC parasite cultures after 24 hr. GO enrichment analysis indicated upregulation of genes in response to xenobiotic stimulus including several phosphatases, hydrolases, and heat shock proteins suggesting stress sensing of the parasites at this stage (*Figure 4c*). GO analyses of downregulated genes, on the other hand, were found to be closely associated with regular metabolic processes and intracellular transport mechanisms suggesting dysregulation of regular cellular activity. However, at this stage, it is probable that the limited depletion of *Pf*MORC (58%) has restricted effects at the transcriptional level (See *Figure 3—figure supplement 1*) and that the observed down-regulated genes are likely due to an indirect effect of cell cycle arrest. At 36 hpi the number of genes that exhibit changes in gene expression were significantly higher with 1319 upregulated and 1150 downregulated (FDR <0.05 and $\log_2$ FC >0.5 or <–0.5) in (-) aTC *Pf*MORC conditions, reflecting of a significant decrease of the protein (~88.2%) and confirming the significant changes observed at the phenotypic level (*Figure 4b*). These results emphasize the cascading effects of *Pf*MORC KD as the parasite progressed throughout its cell cycle. GO analysis indicated an enrichment of upregulated genes involved in multiple pathways including ATP metabolic process, mitochondria, translation, food vacuole, and protein folding (*Figure 4d*); characteristic of not only stress, but also a clear signal of cell cycle arrest in (-) aTC *Pf*MORC samples. Among upregulated genes, we also observed several *var* genes and genes exported at the surface of the red blood cell that could be linked to a significant decreased in *Pf*MORC binding of these gene families as well as a disorganization of the heterochromatin cluster(s) at the trophozoite and schizont stages. GO enrichment analysis for genes that were downregulated included many genes involved in DNA replication, chromosome organization, and mitotic spindle further emphasizing a strong cell cycle arrest and absence of potential major compensatory mechanisms for cell division and chromatin organization (*Figure 4d*). Additional downregulated genes were required for invasion such as several merozoite surface proteins. Our results clearly indicate stress sensing and an arrest in cell cycle progression at the trophozoite and schizont stages (*Supplementary file 6*, *Kiss, 2002*).

Although these results are compelling, the small overlap observed between the ChIP-seq signals and RNA-seq results indicates that major changes observed in gene expression at the schizont stages are not simply the result of reduced *Pf*MORC binding in targeted gene bodies but a combination of direct and indirect effects of the degradation of *Pf*MORC that leads to cell cycle arrest and potential collapse of the heterochromatin (*Figure 4e*).

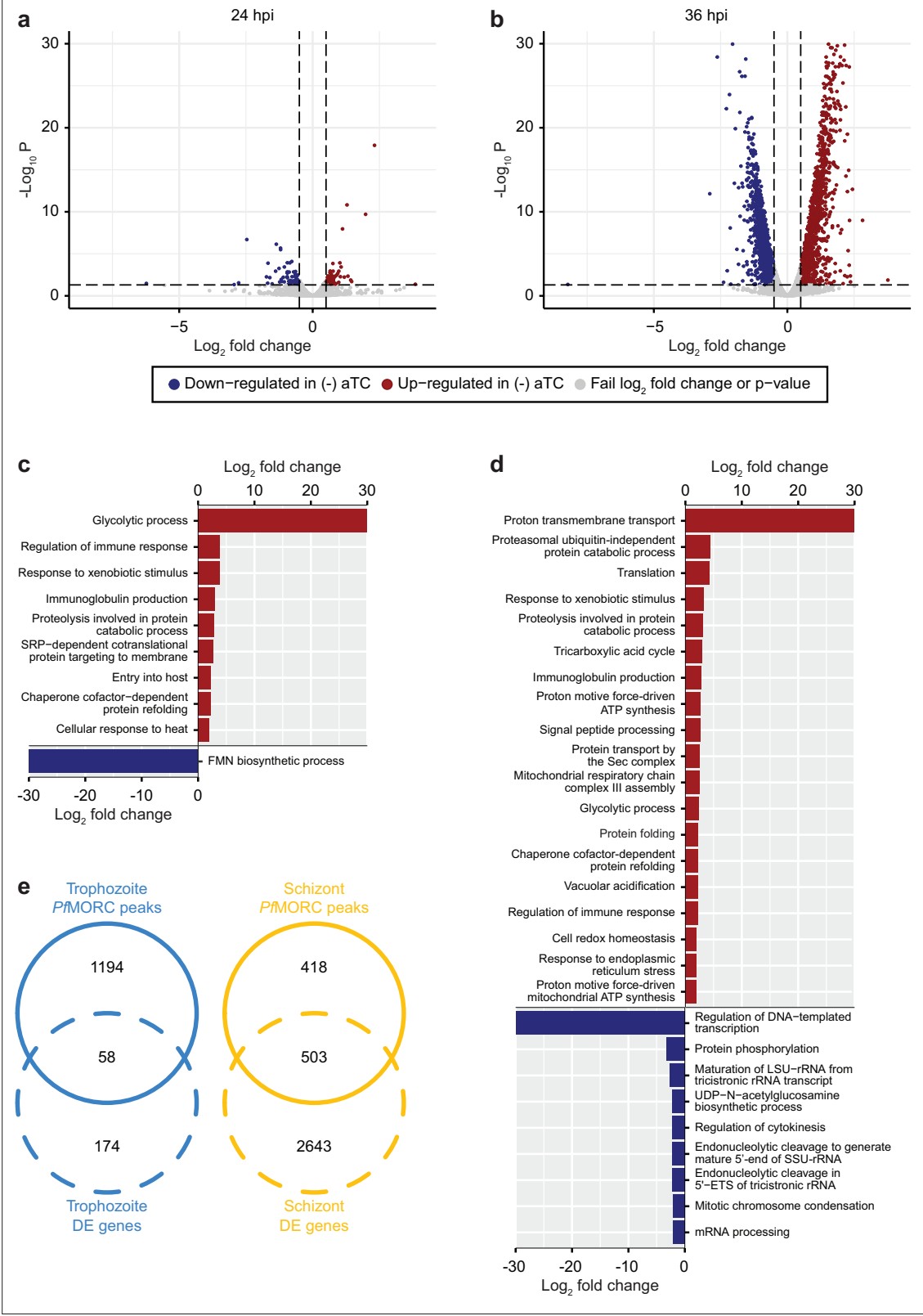

**Figure 4.** *Pf*MORC knockdown (KD) on parasite transcriptome. Volcano plots denoting upregulated (red), and downregulated (blue) genes were discovered through differential expression analysis following *Pf*MORC knockdown at (**a**) 24 hpi (**b**) 36 hpi. Gene ontology enrichment analysis for upregulated (red) and downregulated (blue) genes at (**c**) 24 hpi and (**d**) 36 hpi. (**e**) Overlap of differentially expressed genes and genes containing

*Figure 4 continued on next page*

*Figure 4 continued*

significant peaks called by *Pf*MORC Chromatin immunoprecipitation followed by deep sequencing (ChIP-seq) analysis at the trophozoite and schizont stages.

The online version of this article includes the following figure supplement(s) for figure 4:

**Figure supplement 1.** RNA-seq correlation heatmap.

## *Pf*MORC knockdown erodes antigenic-gene silencing framework

We next sought to analyze the effect of *Pf*MORC down-regulation on the global chromatin landscape. We, therefore, performed a ChIP-seq experiment against histone H3K9me3 and H3K9ac marks in response to *Pf*MORC depletion. Synchronized parasites were equally split between permissive and repressive conditions. At trophozoite (24 hpi) (*Figure 5—figure supplement 1*) and schizont (36 hpi) stages of development, both control (+aTC) and experiment (-aTC) samples were fixed, and parasites collected for chromatin immunoprecipitation followed by sequencing. The experimental procedure was performed in duplicate between (+/-) aTC *Pf*MORC treated lines. Correlation analysis confirms the reproducibility of our ChIP-seq experiments.

Results revealed no significant changes in histone H3K9ac marks across the genome (data not shown) but a reduction in the heterochromatin landscape in the *Pf*MORC-depleted conditions, specifically in the telomere regions of the chromosomes (*Figure 5a*). *Pf*MORC KD resulted in significantly reduced (Mann-Whitney *U* test, p<0.05) H3K9me3 marks within *var* gene promoters from 200 to 800 bp upstream of the TSS, as well as 400–800 bp downstream of the TSS (*Figure 5b–c*). This result coincides with our transcriptomic profile and the upregulation of *var* genes at 36 hpi of (-) aTC samples (*Figure 4d*, *Supplementary file 6*). Further analyses revealed similar trends in other gene families associated with parasite reinvasion and immune response typically silenced under WT conditions. Most noticeable was the reduced H3K9me3 coverage across the *rifin* gene family, which displayed a significant reduction (p<0.05) in all bins from 1000 bp upstream of the TSS to 1000 bp downstream from the end of the gene (*Figure 5b–c*) in response to *Pf*MORC depletion. There is a clear reduction of H3K9me3 coverage of virulence genes shown to be upregulated following *Pf*MORC knockdown (*Figure 5c*). Overall, these findings confirm that *Pf*MORC protein depletion not only affects the chromatin landscape but has a significant impact on heterochromatin. *Pf*MORC downregulation leads to the erosion of heterochromatin integrity most likely required for mutually exclusive gene expression and immune evasion within heterochromatin clusters.

## Knockdown of *Pf*MORC expression results in the loss of tightly regulated heterochromatin structures

To better understand the effect that downregulation of *Pf*MORC has on the chromatin structure and investigate whether changes in chromatin accessibility may explain large changes in gene expression observed using RNA-seq, we performed Hi-C on (+/-) aTC *Pf*MORC cultures at 24 hpi (trophozoite) and 36 hpi (schizont). Biological replicates for each sample were collected and used to generate Hi-C libraries with >37 million reads per replicate/sample, ensuring comprehensive coverage of both intrachromosomal and interchromosomal interactions. After processing (pairing, mapping, and quality filtering) the raw sequences via HiC-Pro (*Servant et al., 2015*), there were approximately 15 million (σ=~10 million) high-quality interaction pairs per sample. Due to the high number of reads and relatively small size of the *P. falciparum* genome (23.3 Mb) compared to higher eukaryotes, we elected to bin our Hi-C data at 10 kb resolution, which allowed the identification of genome-wide patterns while not introducing much noise by binning at too high of a resolution. A high stratum-adjusted correlation (*Figure 6—figure supplement 1*), especially at the 24 hpi time point, suggested that the chromatin structure was consistent between biological replicates, therefore, we chose to combine replicates for downstream analyses and visualization.

Because of variation in the number of valid interaction pairs between (+/-) aTC *Pf*MORC samples, 100 iterations of random sampling were performed on samples with higher read count to obtain ~35 million and ~9 million interaction pairs at 24 hpi and 36 hpi, respectively. ICED normalized intrachromosomal heatmaps displayed a high proportion of interactions at distances less than 10% the length of each chromosome as well as strong subtelomeric interactions and internal regions containing genes involved in antigenic variation (*Figure 6a–b*, *Figure 6—figure supplements 2–5*)

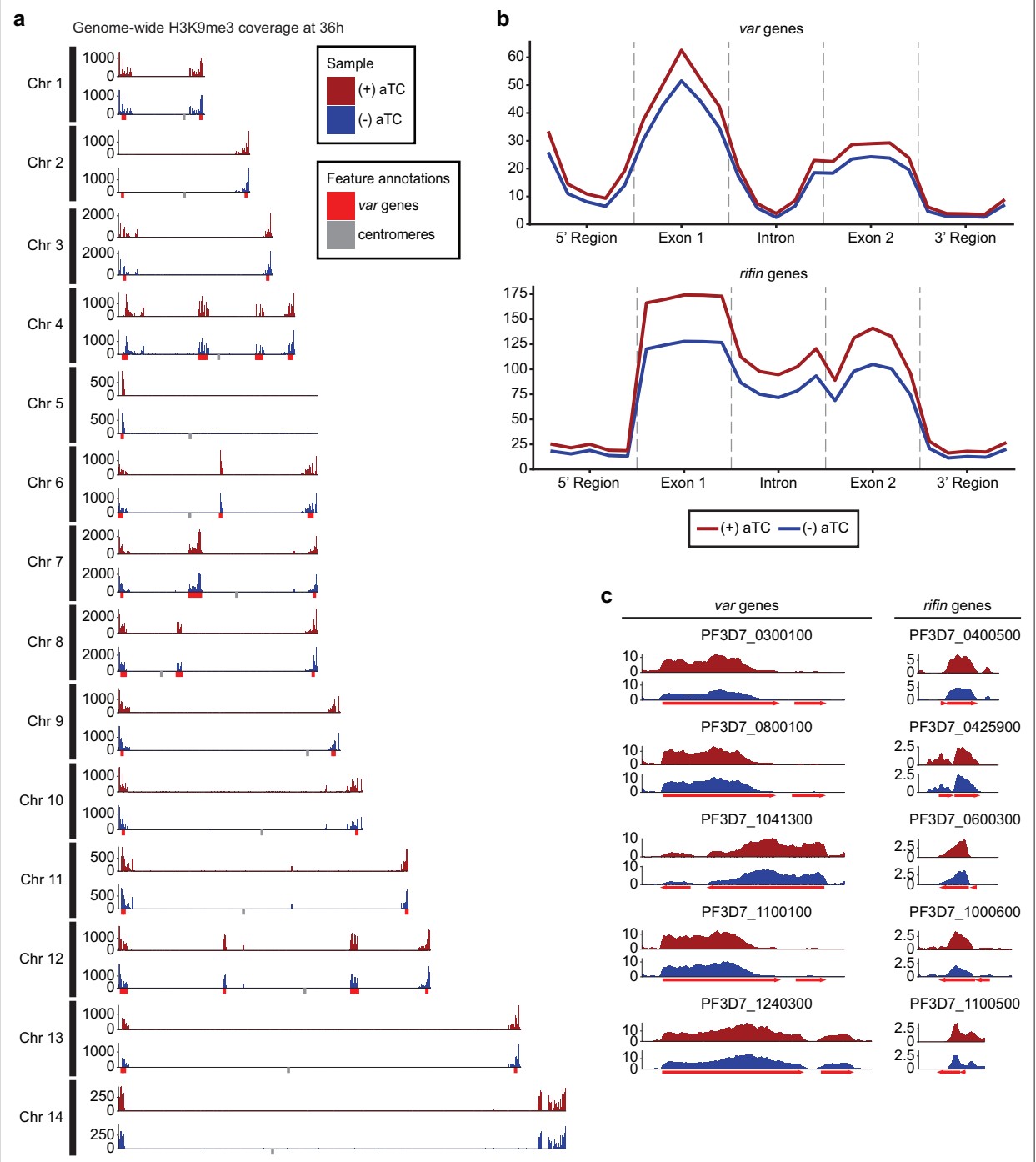

**Figure 5.** Impact of *Pf*MORC knockdown (KD) on heterochromatin markers. (**a**) Genome-wide H3K9me3 coverage of IGG subtracted and per-million normalized (+/-) aTC show similar distribution and concentration within telomeres and antigenic gene clusters highlighted in red. Replicates (n=2) are merged using the mean of the normalized read coverage per base pair. (**b**) Binned coverage of *var* and *rifin* genes from 1 kb upstream of the transcription start site (TSS) to 1 kb downstream of the end. The exons and intron of all genes within these families are split into five equal-sized bins and the 5' and 3' regions are binned into five 200 bp bins. Read counts within each bin are per million and bin length is normalized prior to plotting. (**c**) Coverage of the five most upregulated *var* and *rifin* genes as determined by the transcriptomic analysis which shows elevated coverage in (+) aTC cells.

The online version of this article includes the following figure supplement(s) for figure 5:

**Figure supplement 1.** Impact of 24 hpi *Pf*MORC knockdown (KD) on heterochromatin markers.

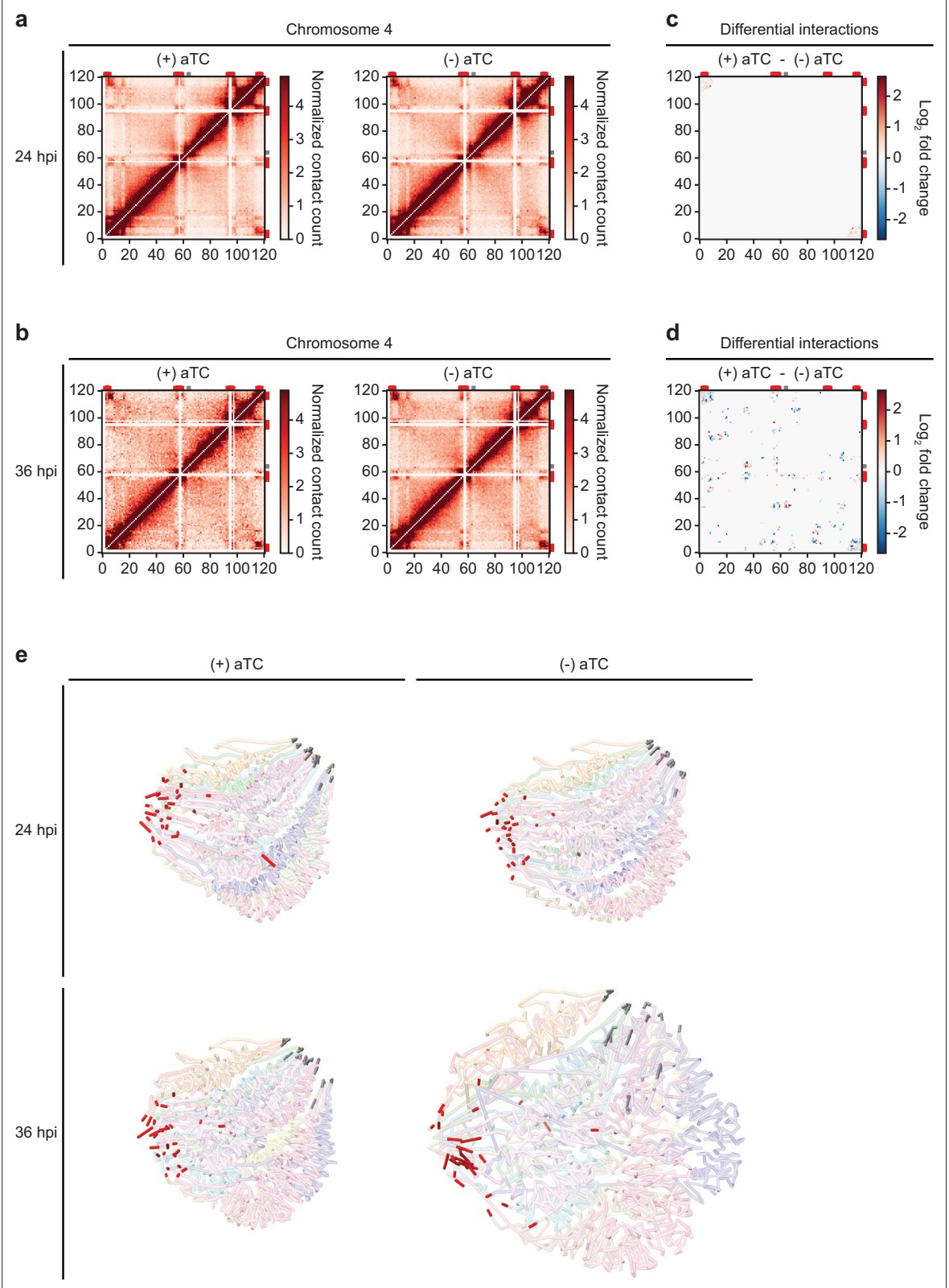

**Figure 6.** Loss of *Pf*MORC expression correlates with heterochromatin expansion. Intrachromosomal interaction heatmaps of (+/-) aTC *Pf*MORC for chromosome 4 at (**a**) 24 hpi and (**b**) 36 hpi displaying heterochromatin clustering within antigenic (*var, rifin,* and *stevor*) gene-dense regions (red). Differential interaction heatmaps highlight changes in chromatin structure following removal of aTC and subsequent *Pf*MORC knockdown at (**c**) 24 hpi and (**d**) 36 hpi. (**e**) Whole-genome 3D models of the chromatin structure at both time points (24 hpi and 36 hpi) and (+/-) aTC.

*Figure 6 continued on next page*

*Figure 6 continued*

The online version of this article includes the following figure supplement(s) for figure 6:

**Figure supplement 1.** Chromatin conformation capture (Hi-C) correlation analyses.

**Figure supplement 2.** Chromatin conformation capture (Hi-C) interaction heatmaps binned at 10 kb resolution.

**Figure supplement 3.** Chromatin conformation capture (Hi-C) interaction heatmaps binned at 10 kb resolution.

**Figure supplement 4.** Chromatin conformation capture (Hi-C) interaction heatmaps binned at 10 kb resolution.

**Figure supplement 5.** Chromatin conformation capture (Hi-C) interaction heatmaps binned at 10 kb resolution.

**Figure supplement 6.** Differential chromatin conformation capture (Hi-C) interaction heatmaps binned at 10 kb resolution.

**Figure supplement 7.** Differential chromatin conformation capture (Hi-C) interaction heatmaps binned at 10 kb resolution.

This pattern was similar to those observed previously at various stages of the life cycle (**Bunnik et al., 2019**; **Servant et al., 2015**; **Bunnik et al., 2018**) and confirm that genes involved in antigenic variation are usually confined within dense heterochromatin rich regions to aid the tight control of *var* genes necessary for their mutually exclusive expression and immune evasion (**Chen et al., 1998**; **Scherf et al., 1998**). Data generated at 36 hpi in (-) aTC *Pf*MORC parasite cultures were, however, tumultuous, with a decrease in defined heterochromatin borders and fewer long-range intrachromosomal interactions across the genome. This may indicate a significant loss of chromatin maintenance in (-) aTC *Pf*MORC parasites (**Figure 6a–b**).

To pinpoint which regions were most strongly affected by the knockdown of *Pf*MORC, we used Selfish (**Ardakany et al., 2019**) to identify differential intrachromosomal and interchromosomal interactions in the (+/-) aTC *Pf*MORC samples. Although the 24 hpi and 36 hpi time points shared similarities in loci most highly affected by *Pf*MORC KD on many chromosomes, the changes at 36 hpi were more significant with a higher $\log_2$ FC (FDR <0.05) at most loci (**Figure 6c–d**, **Figure 6—figure supplements 6–7**). We also observed consistent loss of interactions between most regions containing *var* genes. These results indicate a failed attempt of the (-) aTC *Pf*MORC parasites to maintain their overall chromatin structure with a significant weakening of the tightly controlled heterochromatin cluster.

We further validated our results using PASTIS to generate coordinate matrices and subsequently visualize a consensus three-dimensional model of the chromosome folding and their overall organization within the nucleus (**Varoquaux et al., 2014**). This can indicate changes in spatial distance that best describe the interaction data. The overall change in the chromatin 3D structure was clearly shown by our models of (+/-) aTC *Pf*MORC samples (**Figure 6e**). While the co-localization of centromeres and telomeres in different regions of the nucleus were conserved, the 3D chromatin structure of (-) aTC *Pf*MORC at 36 hpi displayed a clear opening of the chromatin and loss of interactions in most regions. This could explain the large changes in gene expression, including increased expression of all *var* and *rifin* genes in the (-) aTC *Pf*MORC line (**Supplementary file 6**). While heterochromatin maintenance may be essential for tight control of *var* gene expression, preservation of the overall structure of the chromatin may be necessary to regulate the accurate expression of most *P. falciparum* transcripts.

## Discussion

Despite intensive investigations on the dynamic nature of the *Plasmodium* chromatin, progress has been slow in capturing the regulatory factors that maintain and control chromatin structure and gene expression throughout parasite development. Preliminary studies performed by our lab and others have identified *Pf*MORC as one of the most abundant chromatin-bound proteins (**Singh et al., 2021a**; **Zhong et al., 2023**). MORC proteins from plant to animal cells have a broad range of binding sites within the genome and participate in either DNA methylation establishment or heterochromatin formation (**Li et al., 2013**; **Tencer et al., 2020**; **Shao et al., 2010**; **He et al., 2010**; **Zhang et al., 2019**; **Luo et al., 2020**; **Mimura et al., 2010**). They have also been shown to co-localize with TFs in unmethylated promoter regions to alter chromatin accessibility and regulate TF binding and gene expression (**He et al., 2010**; **Zhang et al., 2019**; **Luo et al., 2020**; **Mimura et al., 2010**). For instance, AP2-P (PF3D7_1107800), AP2-G5, ISW1, NUP2, MCM5, and HDAC1 have been shown to interact with MORC in several published works (**Farhat et al., 2020**; **Subudhi et al., 2023**; **Bryant et al., 2020**), including this work. Protein pull-down experiments in *Plasmodium* spp. and *T. gondii* studies have detected MORC in complexes with several AP2 TFs. In *T. gondii*, *Tg*MORC was characterized as

an upstream transcriptional repressor of sexual commitment (*Farhat et al., 2020*; *Zhong et al., 2023*). In *Plasmodium*, the role of *Pf*MORC was still poorly understood. Conflicting results on the essentiality of *Pf*MORC using either KD or KO strategies remained unresolved (*Singh et al., 2021a*; *Zhang et al., 2018*). Using the CRISPR/Cas9 genome editing tool, we have determined the nuclear localization, genome-wide distribution, and regulatory impacts of *Pf*MORC on the parasite chromatin, transcriptome, and cell cycle development. We now have powerful evidence demonstrating that *Pf*MORC is not only critical for parasite cell cycle progression and its survival but also has a direct role in heterochromatin formation and gene silencing, including regulating immune evasion through antigenic variation. Disparities observed in the essentiality of *Pf*MORC for the parasite survival in previous studies (*Singh et al., 2021a*) are most likely the results of weak protein disruption highlighting the need for a significant downregulation of *Pf*MORC for true functional analysis. Using IFA and protein pull-downs, we have confirmed that *Pf*MORC localizes to the nucleus and interacts with multiple ApiAP2 TFs, chromatin remodelers, and epigenetic players associated with heterochromatin regions including H3K9me3, SIP2, HDAC1, and the ISWI chromatin-remodeling complex (SWIB/MDM2) (*Shang et al., 2022*, *Figure 1*). Although some discrepancies have been observed with other studies, mainly due to the fact that MORC protein was not used as the bait protein, our study validated the interaction of MORC with these partners, supporting that *Pf*MORC is in complex with key heterochromatin regulators. Using a combination of ChIP-seq, protein knock down, RNA-seq, and Hi-C experiments, we also demonstrated that the MORC protein is essential for the tight regulation of gene expression. We can speculate that lack of MORC impacts heterochromatin and chromatin compaction, preventing access to gene promoters from TFs and the general transcriptional machinery in a stage-specific manner. Although additional experiments will be required, our hypothesis is reinforced by the fact that downregulation of the *Pf*MORC significantly reduced H3K9me3 coverage in the heterochromatin cluster(s) and 5' flanking regions of *var* and *rifin* genes. While we were unable to confirm a direct role of *Pf*MORC in the sexual conversion due to a nonsense mutation in AP2-G gene in our transfected lines, its strong interaction with AP2-G during the asexual cell cycle indicates that *Pf*MORC in combination with other epigenetic factors may most likely control AP2-G expression and sexual differentiation. It is also important to recognize that the many pathways affected at the transcriptional level throughout the asexual stages in (-) aTC *Pf*MORC lines are not only the direct results of *Pf*MORC downregulation and decrease of its targeted DNA binding site. They are most likely a combination of direct and indirect effects of *Pf*MORC KD warranted by the cell cycle arrest observed at the phenotypic level and the collapse of the chromatin organization confirmed using the chromatin conformation capture experiment. These direct and indirect effects should be carefully considered, and a combination of functional genomic studies should be completed when interpreting changes in gene expression in mutant-generated lines.

All together our work-demonstrated that in addition to its direct role in heterochromatin formation and antigenic variation, *Pf*MORC may act as a potential repressor to control a set of parasite-specific genes including genes involved in parasite egress and invasion, and antigenic variation between the trophozoite to schizont stage transition (*Shang et al., 2022*). As our data confirm the importance of *Pf*MORC and the parasite specificity of several interacting partners, it is tempting to speculate that drugs targeting these protein complexes could lead to novel antiparasitic strategies.

## Materials and methods
### Asexual parasite culture and maintenance
Asexual *P. falciparum* strain NF54 or 3D7 parasites (MRA-1000, MRA-102, respectively) were propagated in 5% of human $O^+$ erythrocytes and 10 mL of RPMI-1640 medium containing 0.5% Albumax II (Invitrogen), 2 mM L-glutamine, 50 mg/L hypoxanthine, 25 mM HEPES, 0.225% $NaHCO_3$ and 10 mg/mL gentamicin. They were maintained at 37 °C and gassed with a sterile mixture of 5% $O_2$, 5% $CO_2$, and 90% $N_2$(*70, 71*). Parasite synchronization was achieved using two 5% D-sorbitol treatments 8 hr apart (*Trager and Jensen, 1976*; *Fidock et al., 1998*).

### Plasmid construction
Flagging of the *P. falciparum* MORC (Pf3D7_1468100) gene spanning position (Ch14: 2785386–2797934 (-)) was performed using a two-plasmid design to insert a 3 x HA tag. The pCasG-Cas9-sgRNA

plasmid vector (gifted by Dr. Sean Prigge) contains the site to express the sgRNA, along with the yDHODH gene as the positive selection marker. The gRNA oligos (*Supplementary file 1*) were ligated after digestion of the plasmid with BsaI. The pDC2-cam-Cas9-U6 plasmid (gifted by Dr. Marcus Lee) was digested with BamHI and ApaI to remove the eGFP tag from the backbone. 453 bp of the C-terminal region of *Pf*MORC and 469 bp of 3'UTR were amplified from *P. falciparum* genomic DNA with their respective primers (*Supplementary file 1*). A 3 x HA-tag was fused to the C-terminal region of the amplified product along with the 3'UTR formed through the Gibson assembly master mix (NEB, E2611S). To generate the *Pf*MORC-HA knockdown constructs, a pKD$^{PfAUBL}$ plasmid (gifted by Dr. Sean Prigge) (*Rajaram et al., 2020*) was digested with AscI and AatII. Homology arms HA1 and HA2 of *Pf*MORC were amplified with respective primers (*Supplementary file 1*) having 20 bp overhang and inserted into the digested plasmid using a Gibson assembly mix. The resulting pKD-MORC-HA-TetR-DOZI construct was linearized with the EcoRV enzyme prior to transfection. All constructs were confirmed through restriction enzyme digestion and Sanger sequencing.

Plasmids were isolated from 250 mL cultures of *Escherichia coli* (XL10-Gold Ultracompetent Cells, Agilent Cat. 200314) and 60 μg of pDC2-MORC-HA or linearized pKD-MORC-HA-TetR-DOZI, were used with 60 μg of pCasG-plasmid containing gRNA to transfect 200 μl of fresh red blood cells (RBCs) infected with 3–5% early ring stage parasites. After one erythrocytic cycle, transfected cultures were supplemented with 1.5 μM WR99210 (2.6 nM) (provided by the Jacobus Pharmaceuticals, Princeton, NJ) and 2.5 μg/mL Blasticidin (RPI Corp B12150-0.1). *Pf*MORC-HA-TetR-DOZI transfected parasites were maintained with 500 nM anhydrotetracycline (aTC) (*Ganesan et al., 2016*; *Nasamu et al., 2021*). Media and drug selection was replenished every 24 hr for seven consecutive days after which DSM-1 drug selection was halted Once parasites were detected by microscopy, integration of the insert was confirmed by PCR amplification. To generate genetically homogenous parasite lines, the transfected parasites were serially diluted to approximately 0.5% parasite/well, into 96 well plates.

## Molecular analysis of the transgenic lines

Genomic DNA (gDNA) was extracted and purified using the DNeasy Blood & Tissue kit (Qiagen) following instructions from the manufacturer. The diagnostic PCR analysis was used to genotype the transfected lines using the primers listed in Supplementary data 1. The PCR amplification was conducted using KAPA HiFi HotStart ReadyMix (Roche) and amplicons were analyzed by gel electrophoresis followed by sequencing. For whole genome sequencing, genomic DNA was fragmented using a Covaris S220 ultrasonicator and libraries were generated using the KAPA LTP Library Preparation Kit (Roche, KK8230). To verify that the insertion was present in the genome at the correct location in both transfected lines, reads were mapped using Bowtie2 (v2.4.4) to the *P. falciparum* 3D7 reference genome (PlasmoDB, v48), edited to include the insertion sequence in the intended location. Integrative Genomic Viewer (IGV, Broad Institute) was used to verify that reads aligned to the modified sequence.

## Variant analysis by genome-wide sequencing

To call variants (SNPs/indels) in the transfected lines compared to a previously sequenced control 3D7 line, genomic DNA reads were first trimmed to adapters and aligned to the *Homo sapiens* genome (assembly GRCh38) to remove human-mapped reads. Remaining reads were aligned to the *P. falciparum* 3D7 genome using bwa (version 0.7.17) and PCR duplicates were removed using PicardTools (Broad Institute). GATK HaplotypeCaller (https://gatk.broadinstitute.org/hc/en-us) was used to call variants between the sample and the 3D7 reference genome for both the transfected lines and the NF54 control. Only variants that were present in both transfected lines but not the NF54 control line were kept. We examined only coding-region variants and removed those that were synonymous variants or were located in *var*, *rifin*, or *stevor* genes. Quality control of variants was done by hard filtering using GATK guidelines.

## Immunofluorescence assays

A double-staining immunofluorescence assay was used on NF54 control and transgenic parasite lines of mixed parasite population growth stages. Parasites were washed in incomplete medium prior to fixing onto coverslips with 4% paraformaldehyde for 20 min at RT under darkness. After fixation, samples were washed three to five times with 1 x PBS followed by permeabilization with 0.5% Triton

X-100 in PBS for 25 min at RT. Subsequently, samples were subjected to PBS washes and then incubated overnight at 4 °C in blocking buffer (2 mg/ml BSA solution in PBS containing 0.05% Tween-20 solution). Following overnight blocking, samples were washed and incubated for 1 hr at RT with Anti-HA Rb Ab (1:500, Abcam, ab9110) in a blocking buffer. After primary Ab incubation, samples were subject to 3 x washes with wash buffer (1 x PBS containing 0.05% Tween-20) followed by incubation with anti-rabbit DyLight 550 (Abcam ab98489; 1:500) secondary antibody for 1 hr at room temperature. After incubation and a series of washes with wash buffer, slides were incubated with anti-H3K9me3 antibody, Alexa Fluor 488 conjugate (Millipore 07–442-AF488; 1:100) for 1 hr at RT. Slides are then washed and mounted in Vectashield Antifade Mounting Medium with DAPI (Vector Laboratories, H-1200). Images were acquired using a Keyence BZ-X810 Fluorescence Microscope and were processed through ImageJ.

## Western blotting

Parasites were synchronized twice at 8 hr intervals between synchronizations. After one cycle, culture was washed 3 x with complete media to remove residual aTC followed by dividing the cultures into two equal conditions. Parasites were grown with or without 500 nM aTC for 24 and 36 hr. after which RBCs were lysed using 0.15% saponin and parasites were collected after being washed with ice-cold 1 X PBS. Proteins were recovered from the lysed parasites after 30 min of incubation in lysis buffer (150 mM NaCl, 0.5 % NP40, 50 mM Tris pH 8, 1 mM EDTA, and protease inhibitors) and 10 s of sonication. Proteins were quantified with Pierce BCA Protein Assay Kit (Thermo Fisher 23227). 20 µg of proteins were loaded onto 3–8% Criterion XT Tris-Acetate Midi Protein Gels (Bio-Rad, 3450129). After migration, proteins were transferred onto a PVDF membrane and the membrane was blocked and then probed overnight with an anti-HA tag antibody (1:2500, Abcam, ab9110) as well as anti-aldolase antibody as a loading control (1:10,000, abcam, ab252953). After primary Ab incubation, blots were subsequently washed 3 x with a washing buffer followed by HRP-labeled Goat anti-Rabbit IgG (H + L) (1:10,000, Novex, A16104). Clarity Western ECL Substrate (Bio-Rad, 1705060) was applied to reveal the blots. Relative abundance of (+/-) aTC *Pf*MORC was calculated by Bio-Rad ChemiDoc image lab software.

## Immunoprecipitation followed by MudPIT mass spectrometry

Mid- to late-stage asexual parasites were collected following saponin treatment and purified samples were then resuspended into fresh IP buffer (50 mM Tris-HCl pH 7.5, 300 mM NaCl, 0.5 mM EDTA, 0.5 mM EGTA, 2 mM AEBSF 0.5% Triton X-100, and EDTA-free protease inhibitor cocktail (Roche)). Post-cell lysis solution was homogenized via sonication for 6–9 rounds. The soluble extracts were centrifuged at 13,000 x g for 15 min at 4 °C. The lysates were precleared with Dynabeads Protein A (Invitrogen) for 1 hr at 4 °C. Anti-HA tag antibodies (1:2500, Abcam, ab9110) are added to control and HA-tagged *Pf*MORC precleared protein extract samples for 1 hr at 4 °C under constant rotation followed by the addition of fresh Dynabeads Protein A beads to each sample and incubated overnight at 4 °C. Dynabeads were washed three times with 500 µL of buffer (PBS, 0.05% Tween-20). Proteins were eluted into an elution buffer (50 mM Tris-HCl pH 6.7, 100 mM DTT, and 2% SDS). The eluent was subsequently precipitated overnight in 20% TCA followed by cold acetone washes. The urea-denatured, reduced, alkylated, and digested proteins were analyzed by Multidimensional Protein Identification Technology (MudPIT) on an Orbitrap Elite mass spectrometer coupled to an Agilent 1260 series HPLC, as described previously (*Florens and Washburn, 2006*).

## Proteomics data processing and analysis

Tandem mass (MS/MS) spectra were interpreted using ProluCID v.1.3.3 (*Xu et al., 2015*) against a database consisting of 5527 non-redundant (NR) *Plasmodium falciparum* 3D7 proteins (PlasmoDB, v42), 36661 NR human proteins (NCBI, 2018-03-30 release), 419 common contaminants (human keratins, IgGs, and proteolytic enzymes), together with shuffled versions of all of these sequences. DTASelect v.1.9 (*Tabb et al., 2002*) and swallow v.0.0.1, an in-house developed software (https://github.com/tzw-wen/kite copy archived at *Wen, 2024*) were used to control FDRs resulting in protein FDRs less than 1.86%. All datasets were contrasted against their merged data set, respectively, using Contrast v1.9 (*Tabb et al., 2002*) and in-house developed sandmartin v.0.0.1 (https://github.com/tzw-wen/kite/tree/master/kitelinux). Our in-house developed software, NSAF7 v.0.0.1 (https://github.

com/tzw-wen/kite/tree/master/windowsapp/NSAF7x64), was used to generate spectral count-based label-free quantitation results (*Zhang et al., 2010*). QPROT (*Choi et al., 2015*; *Choi et al., 2008*) was used to calculate values of $\log_2$ fold change and Z-statistic to compare two replicate *Pf*MORC affinity purifications to two negative controls. Proteins enriched in the *Pf*MORC-HA samples with values of $\log_2$ fold change ≥2 and Z-statistic ≥5 were considered significantly enriched.

## ChIP assay

*Pf*MORC-HA and *Pf*MORC KD asexual stage parasites were supplemented with and without aTC along with NF54 parasites (as a control) and were harvested at 24 and 36 HPS and cross-linked with 1% formaldehyde for 10 min at 37 °C followed by quenching with 150 mM glycine and 3 x washing with 1 x PBS. The pellets were resuspended in 1 mL of nuclear extraction buffer (10 mM HEPES, 10 mM KCl, 0.1 mM EDTA, 0.1 mM EGTA, 1 mM DTT, 0.5 mM AEBSF, and 1 x Protease inhibitor cocktail), and incubated on ice for 30 min before addition of NP-40/Igepal to a final of 0.25%. After lysis, the parasites are subject to homogenization by passing through a 26 G 3/8 needle/syringe to burst the nucleus. After centrifugation for 20 min at 2,500 x g at 4 °C, samples were resuspended in 130 µL of shearing buffer (0.1% SDS, 1 mM EDTA, 10 mM Tris-HCl pH 7.5 and 1 X Protease inhibitor cocktail) and transferred to a 130 µL Covaris sonication microtube. The samples were then sonicated using a Covaris S220 Ultrasonicator for 8 min (Duty cycle: 5%, intensity peak power: 140, cycles per burst: 200, bath temperature: 6 °C). The samples were transferred to ChIP dilution buffer (30 mM Tris-HCl pH 8.0, 3 mM EDTA, 0.1% SDS, 30 mM NaCl, 1.8% Triton X-100, 1 X protease inhibitor tablet, 1 X phosphatase inhibitor tablet) and centrifuged for 10 min at 13,000 rpm at 4 °C, retaining the supernatant. For each sample, 13 µL of protein A agarose/salmon sperm DNA beads were washed three times with 500 µL ChIP dilution buffer by centrifuging for 1 min at 1000 rpm at room temperature. For pre-clearing, the diluted chromatin samples were added to the beads and incubated for 1 hr at 4 °C with rotation, then pelleted by centrifugation for 1 min at 1000 rpm. Before adding antibody,~10% of each sample was taken as input. 2 µg of anti-HA tag (1:2500, Abcam, ab9110) or rabbit polyclonal anti-H3K9me3 or anti-H3K9ac (Millipore no. 07–442 and H0913) antibodies were added to the sample and incubated overnight at 4 °C with rotation. For each sample, rabbit IgG antibody (Cat.689 No. 10500 c, Invitrogen) was used as a negative-control library. Per sample, 25 µL of protein A agarose/salmon sperm DNA beads were washed with ChIP dilution buffer (no inhibitors), blocked with 1 mg/mL BSA for 1 hr at 4 °C, and then washed three more times with buffer. 25 µL of washed and blocked beads were added to the sample and incubated for 1 hr at 4 °C with continuous mixing to collect the antibody/protein complex. Beads were pelleted by centrifugation for 1 min at 1000 rpm at 4 °C. The bead/antibody/protein complex was then washed with rotation using 1 mL of each buffer twice; low salt immune complex wash buffer (1% SDS, 1% Triton X-100, 2 mM EDTA, 20 mM Tris-HCl pH 8.0, 150 mM NaCl), high salt immune complex wash buffer (1% SDS, 1% Triton X-100, 2 mM EDTA, 20 mM Tris-HCl pH 8.0, 500 mM NaCl), and TE wash buffer (10 mM Tris-HCl pH 8.0, 1 mM EDTA). Complexes were eluted from antibodies by adding 250 µL of freshly prepared elution buffer (1% SDS, 0.1 M sodium bicarbonate). 5 M NaCl were added to the eluate, and cross-linking was reversed by heating at 45 °C overnight followed by the addition of 15 µL of 20 mg/mL RNase A with 30 min incubation at 37 °C. After this, 10 µL of 0.5 M EDTA, 20 µL of 1 M Tris-HCl pH 7.5, and 2 µL of 20 mg/mL proteinase K were added to the eluate and incubated for 2 hr at 45 °C. DNA was recovered by phenol/chloroform extraction and ethanol precipitation. DNA was purified using Agencourt AMPure XP beads. Libraries were then prepared from this DNA using the KAPA LTP Library Preparation Kit (Roche, KK8230) and sequenced on a NovaSeq 6000 machine.

## ChIP-seq analysis

FastQC (version 0.11.8, https://www.bioinformatics.babraham.ac.uk/projects/fastqc/) was used to analyze raw read quality. Any adapter sequences were removed using Trimmomatic (version 0.39, http://www.usadellab.org/cms/?page=trimmomatic). Bases with Phred quality scores below 20 were trimmed using Sickle (version 1.33, https://github.com/najoshi/sickle; *najoshi, 2015*). The resulting reads were mapped against the *P. falciparum* genome (v48) using Bowtie2 (version 2.4.4). Using Samtools (version 1.11), only properly paired reads with mapping quality 40 or higher were retained, and reads marked as PCR duplicates were removed by PicardTools MarkDuplicates (version 2.18.0, Broad Institute). Genome-wide read counts per nucleotide were normalized by dividing millions of

mapped reads for each sample (for all samples including input) and subtracting IgG read counts from the anti-HA IP counts. Peak calling was performed using MACS3 (version 3.0.0a7) (*Zhang et al., 2008*) with the upper and lower limits for fold enrichment set to 2 and 50, respectively.

## Phenotypic analyses

*P. falciparum* NF54 line along with transgenic *Pf*MORC-HA-TetR-DOZI line were synchronized, grown for one cycle post-synchronization, and subject to three consecutive washes with 1 X PBS before being supplemented either with or without (aTC) for 48 hr or 72 hr (depending on which stage the parasites were washed). Culture media were replaced daily with or without aTC supplementation.

## Quantitative growth assay

Synchronous cultures either of ring or trophozoite stage at 0.5% parasitemia were grown with or without aTC (500 nM) into a 96-well plate in triplicates for each time point. Parasites were collected at every 24 hr time for three cycles and subjected to SYBR green assay (Thermo Fisher, S7523).

## RNA-seq library preparation

Parasites at the ring, trophozoite, or schizont stage were extracted following saponin treatment before flash freezing. Two independent biological replicates were generated for each time point, culture condition, and line. Total RNA was extracted with TRIzol LS Reagent (Invitrogen) followed by incubation for 1 hr with 4 units of DNase I (NEB) at 37 °C. RNA samples were visualized by RNA electrophoresis and quantified on Synergy HT (BioTek). mRNA was then purified using NEBNext Poly(A) mRNA Magnetic Isolation Module (NEB, E7490) according to the manufacturer's instructions. Libraries were prepared using NEBNext Ultra Directional RNA Library Prep Kit (NEB, E7420) and amplified by PCR with KAPA HiFi HotStart Ready Mix (Roche). PCR conditions consisted of 15 min at 37 °C followed by 12 cycles of 98 °C for 30 s, 55 °C for 10 s, and 62 °C for 1 min 15 s, then finally one cycle for 5 min at 62 °C. The quantity and quality of the final libraries were assessed using a Bioanalyzer (Agilent Technology Inc). Libraries were sequenced using a NovaSeq 6000 DNA sequencer (Illumina), producing paired-end 100 bp reads.

## RNA-seq data processing and differential expression analysis

FastQC [https://www.bioinformatics.babraham.ac.uk/projects/fastqc/] was used to analyze raw read quality and thus 11 bp of each read and any adapter sequences were removed using Trimmomatic (v0.39) [http://www.usadellab.org/cms/?page=trimmomatic]. Bases were trimmed from reads using Sickle with a Phred quality threshold of 25 (v1.33) [https://github.com/najoshi/sickle] and reads shorter than 18 bp were removed. The resulting reads were mapped against the *P. falciparum* 3D7 genome (PlasmoDB, v53) using HISAT2 (v2-2.2.1) with default parameters. Uniquely mapped, properly paired reads with a mapping quality of 40 or higher were retained using SAMtools (v1.11) [http://samtools.sourceforge.net/]. Raw read counts were determined for each gene in the *P. falciparum* genome using BedTools [https://bedtools.readthedocs.io/en/latest/#] to intersect the aligned reads with the genome annotation. Differential expression analysis was performed using DESeq2 to call up- and down-regulated genes (FDR<0.05 and $\log_2$ FC>0.5). Volcano plots were made using the R package EnhancedVolcano.

## Hi-C library preparation

(+/-) aTC parasites at 24 hpi and 36 hpi were cross-linked using 1.25% formaldehyde for 15 min at 37 °C in 10 mL total volume followed by quenching with 150 mM glycine. Parasites were then washed three times with chilled 1 x PBS on a rocking platform. Parasite nuclei were released using lysis buffer (10 mM Tris-HCl, pH 8.0, 10 mM NaCl, 2 mM AEBSF, 0.25% Igepal CA-630, and 1 X EDTA-free protease inhibitor cocktail (Roche)) and 15 syringe passages through a 26.5-gauge needle. After releasing the crosslinked chromatin with 0.1% SDS, chromatin was digested using MboI restriction enzyme overnight at 37 °C. Thereafter Hi-C libraries were prepared as previously described (*Bunnik et al., 2019*; *Gupta et al., 2021*).

## Hi-C data processing, differential interaction analysis, and generation of 3D models

Paired-end Hi-C libraries were processed (pairing, mapping, quality filtering, binning, and normalization) using HiC-Pro (*Servant et al., 2015*). A mapping quality cutoff of 30 was set while aligning

to the *P. falciparum* genome (PlasmoDB, v58), and the resulting reads were binned at 10 kb resolution and ICED-normalized. Stratum-adjusted correlation coefficients were calculated using HiCRep (*Yang et al., 2017*), then replicates were merged to generate a single consensus sample for each time point and condition. The normalized matrices were per-million read count normalized and maximum values for the heatmap color scale on each chromosome was set to the maximum value for all samples to allow for direct comparisons between each condition and time point. Furthermore, due to the overrepresentation of short-range interactions, the maximum values for each heatmap were also set to the 90th percentile of each chromosome to slightly compress highly interacting regions and enhance the visualization of interactions. Significant intrachromosomal interactions were identified with FitHiC (*Ay et al., 2014*) to identify the log-linear relationship between contact probability and genomic distance. Differential interchromosomal and intrachromosomal interactions between (+) aTC and (-) aTC were identified using Selfish (*Ardakany et al., 2019*) with default parameters (FDR <0.05). The maximum and minimum values for the color scales in the differential heatmaps were set to the absolute value of the largest $\log_2$ fold change within each chromosome. PASTIS (*Varoquaux et al., 2014*) was used to generate coordinate matrices from the raw read count matrices output by HiC-Pro, and then visualized as 3D chromatin models in ChimeraX (*Goddard et al., 2018*), highlighting regions containing *var* genes, telomeres, and centromeres.

## Statistical analyses

Parasitemia and the proportion of asexual stages were analyzed using a two-way ANOVA with Tukey's test for multiple comparisons. Significant differences were indicated as follows: *$p<0.05$; **$p<0.01$, ***$p<0.001$, and ****$p<0.0001$. Statistical tests were performed with GraphPad Prism. Figures were generated with GraphPad Prism and BioRender.

## Acknowledgements

This work was supported by NIH grants to KGLR (nos. 1R01 AI136511 and R21 AI142506-01) and by the University of California, Riverside to KGLR (no. NIFA-Hatch-225935). This publication includes data generated at the UC San Diego IGM Genomics Center utilizing an Illumina NovaSeq 6000 that was purchased with funding from a National Institutes of Health SIG grant (#S10 OD026929).

## Additional information

### Funding

| Funder | Grant reference number | Author |
| --- | --- | --- |
| National Institute of Allergy and Infectious Diseases | R01 AI136511 | Karine G Le Roch |
| National Institute of Allergy and Infectious Diseases | R21 AI142506 | Karine G Le Roch |
| University of California, Riverside | NIFA-Hatch-225935 | Karine G Le Roch |

The funders had no role in study design, data collection and interpretation, or the decision to submit the work for publication.

### Author contributions

Zeinab M Chahine, Data curation, Formal analysis, Investigation, Methodology, Writing – original draft; Mohit Gupta, Data curation, Methodology, Writing – review and editing; Todd Lenz, Steven Abel, Formal analysis, Writing – review and editing; Thomas Hollin, Data curation, Formal analysis, Methodology, Writing – original draft, Writing – review and editing; Charles Banks, Anita Saraf, Formal analysis, Methodology, Writing – review and editing; Jacques Prudhomme, Data curation, Formal analysis, Methodology, Writing – review and editing; Suhani Bhanvadia, Formal analysis; Laurence A Florens, Formal analysis, Methodology, Project administration, Writing – review and editing; Karine G

Le Roch, Conceptualization, Formal analysis, Supervision, Funding acquisition, Writing – original draft, Project administration

### Author ORCIDs
Zeinab M Chahine ⓘ https://orcid.org/0000-0001-8208-5869
Thomas Hollin ⓘ https://orcid.org/0000-0002-8089-3253
Anita Saraf ⓘ https://orcid.org/0000-0001-6756-0975
Jacques Prudhomme ⓘ https://orcid.org/0000-0002-6161-5194
Laurence A Florens ⓘ https://orcid.org/0000-0002-9310-6650
Karine G Le Roch ⓘ https://orcid.org/0000-0002-4862-9292

Reviewer #1 (Public review): https://doi.org/10.7554/eLife.92499.3.sa1
Author response https://doi.org/10.7554/eLife.92499.3.sa2

---

## Additional files

### Supplementary files
• Supplementary file 1. *Pf*MORC plasmid constructs to generate transgenic lines. All constructs were performed through listed primers, restriction sites, and gRNAs.

• Supplementary file 2. Mapped whole genome sequencing results of *Pf*MORC transfectants.

• Supplementary file 3. MORC-HA associated proteins identified via MudPIT analysis. (a) Proteins identified by MudPIT analysis after MORC-HA immunoprecipitation. *Pf*MORC (PF3D7_1468100) is highlighted in yellow and significantly purified proteins are in red. The filters used are QPROT $\log_2$ fold change >2 and Z statistic >5. Detailed protein list is provided in an additional sheet. (b) Raw data showing proteins identified by MudPIT analysis after *Pf*MORC-HA immunoprecipitation.

• Supplementary file 4. Peak calling the result of *Pf*MORC Chromatin immunoprecipitation followed by deep sequencing (ChIP-seq). Results were obtained at (a) ring stages, (b) trophozoite stages, and (c) schizont stages of cell progression.

• Supplementary file 5. Parasite Survival Assay illustrating the effect of *Pf*MORC down-regulation on parasites at the ring stage (top) or trophozoite stage (bottom) cell cycle. Experiments were conducted in triplicates with parasitemia quantified via relative fluorescence units (RFUs) obtained by SYBR green assay for each time point. Significance derived through two-way ANOVA.

• Supplementary file 6. Effect of PfMORC KD on transcriptomic profile at 24 and 36 hr. (a) RNA-seq read counts of *Pf*MORC at 24 and 36 hr. DEseq2 analysis of differentially expressed genes of *Pf*MORC at (b) 24 hr and (c) 36 hr.

• MDAR checklist

• Figure 3—figure supplement 1—source data 1. Original, uncropped gel images and western blot files for displayed PfMORC-HA and PfMORC-HA-TetR-DOZI results in *Figure 3—figure supplement 1*.

• Figure 3—figure supplement 1—source data 2. Uncropped and marked gel images and western blot files for displayed PfMORC-HA and PfMORC-HA-TetR-DOZI results in *Figure 3—figure supplement 1*.

### Data availability
Sequence reads for all sequencing experiments have been deposited in the NCBI Sequence Read Archive with accession PRJNA994684. Original data underlying this manuscript generated at the Stowers Institute can be accessed from the Stowers Original Data Repository at http://www.stowers.org/research/publications/LIBPB-2404. The mass spectrometry dataset generated for this study is available from the MassIVE data repository using the identifier MSV000092353. All other data are available in the main text, methods, and supplementary data.

The following datasets were generated:

| Author(s) | Year | Dataset title | Dataset URL | Database and Identifier |
|---|---|---|---|---|
| Lab LR | 2023 | PfMORC protein regulates chromatin accessibility and transcriptional repression in the human malaria parasite, *P. falciparum* | https://www.ncbi.nlm. nih.gov/bioproject/ PRJNA994684/ | NCBI BioProject, PRJNA994684 |
| Roch F | 2023 | PfMORC protein regulates chromatin accessibility and transcriptional repression in the human malaria parasite, *P. falciparum* | https://massive. ucsd.edu/ ProteoSAFe/dataset. jsp?accession= MSV000092353 | Massive, MSV000092353 |
| Banks C, Saraf A, Florens L | 2024 | Index of /publications/ LIBPB-2404 | https://odr.stowers. org/publications/ LIBPB-2404/ | Stowers Institute, LIBPB-2404 |

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
