## [Editor Report · eLife Assessment]

This **important** study examines the role of Microrchidia (MORC) proteins in the human malaria parasite *Plasmodium falciparum*. **Solid** experimental results, including genome editing and chromatin profiling methods (ChIP-seq and Hi-C), provide a comprehensive picture of the critical role MORC plays in shaping parasite chromatin. Depletion of MORC results in a lethal collapse of heterochromatin and parasite death, nominating the factor as a new target of antimalarial therapies.

---

## [Referee Report · Reviewer #1 (Public review)]

Summary:

The authors investigated the function of Microrchidia (MORC) proteins in the human malaria parasite *Plasmodium falciparum*. Recognizing MORC's implication in DNA compaction and gene silencing across diverse species, the study aimed to explore the influence of PfMORC on transcriptional regulation, life cycle progression and survival of the malaria parasite. Depletion of PfMORC leads to the collapse of heterochromatin and thus to the killing of the parasite. The potential regulatory role of PfMORC in the survival of the parasite suggests that it may be central to the development of new antimalarial strategies.

Strengths:

The application of the cutting-edge CRISPR/Cas9 genome editing tool, combined with other molecular and genomic approaches, provides a robust methodology. Comprehensive ChIP-seq experiments indicate PfMORC's interaction with sub-telomeric areas and genes tied to antigenic variation, suggesting its pivotal role in stage transition. The incorporation of Hi-C studies is noteworthy, enabling the visualization of changes in chromatin conformation in response to PfMORC knockdown.

Weaknesses:

Although disruption of PfMORC affects chromatin architecture and stage-specific gene expression, determining a direct cause-effect relationship requires further investigation. Furthermore, while numerous interacting partners have been identified, their validation is critical and understanding their role in directing MORC to its targets or in influencing the chromatin compaction activities of MORC is essential for further clarification. In addition, the authors should adjust their conclusions in the manuscript to more accurately represent the multifaceted functions of MORC in the parasite.

---

## [Author Response]

The following is the authors’ response to the original reviews.

**Reviewer #1 (Public Review):**
Summary: The authors investigated the function of Microrchidia (MORC) proteins in the human malaria parasite *Plasmodium falciparum*. Recognizing MORC's implication in DNA compaction and gene silencing across diverse species, the study aimed to explore the influence of PfMORC on transcriptional regulation, life cycle progression and survival of the malaria parasite. Depletion of PfMORC leads to the collapse of heterochromatin and thus to the killing of the parasite. The potential regulatory role of PfMORC in the survival of the parasite suggests that it may be central to the development of new antimalarial strategies.Strengths: The application of the cutting-edge CRISPR/Cas9 genome editing tool, combined with other molecular and genomic approaches, provides a robust methodology. Comprehensive ChIP-seq experiments indicate PfMORC's interaction with sub-telomeric areas and genes tied to antigenic variation, suggesting its pivotal role in stage transition. The incorporation of Hi-C studies is noteworthy, enabling the visualization of changes in chromatin conformation in response to PfMORC knockdown.

We greatly appreciate the overall positive feedback and cognisense of our efforts. Our application of CRISPR/Cas9 genome editing tools coupled with complementary cellular and functional approaches shed light on the importance of _Pf_MORC in maintaining chromatin structural integrity in the parasite and highlights this protein as a promising target for novel therapeutic intervention.

Weaknesses: Although disruption of PfMORC affects chromatin architecture and stage-specific gene expression, determining a direct cause-effect relationship requires further investigation.

Our conclusions were made on the basis of multiple, unbiased molecular and functional genomic assays that point to the relevance of the _Pf_MORC protein in maintaining the parasite’s chromatin landscape. Although we do not claim to have precise evidence on the step-by-step pathway to which _Pf_MORC is involved, we bring forth first-hand evidence of its role in heterochromatin binding, gene-regulation and its association with major TFs as well as chromatin remodeling and modifying enzymes. We however agree with the comment regarding the lack of direct effects of _Pf_MORC KD and have since provided additional evidence by performing ChIP-seq experiments against H3K9me3 and H3K9ac during KD. Our new results are presented in Fig. 5. We showed that the level of H3K9me3 decreased significantly during _Pf_MORC KD.

Furthermore, while numerous interacting partners have been identified, their validation is critical and understanding their role in directing MORC to its targets or in influencing the chromatin compaction activities of MORC is essential for further clarification. In addition, the authors should adjust their conclusions in the manuscript to more accurately represent the multifaceted functions of MORC in the parasite.

Validation of the identified interacting partners is indeed critical and essential to understanding their role in directing MORC to its targets. Our protein pull down experiments have been done using several biological replicates. Several of the interacting partners have also been identified and published by other labs and collaborators. To confirm our results, we completed a direct comparison of our work with previous published work. Results have now been incorporated into the revised manuscript to confirm the identified interacting partners and the accuracy of the data we obtained in our experiment. Molecular validation of novel proteins identified in our protein pull down requires generation of tagged lines and may take a few more years but will be submitted for publication in a follow up manuscript.

**Reviewer #2 (Public Review):**
Summary: This paper, titled "Regulation of Chromatin Accessibility and Transcriptional Repression by PfMORC Protein in *Plasmodium falciparum*," delves into the PfMORC protein's role during the intra-erythrocytic cycle of the malaria parasite, *P. falciparum*. Le Roch et al. examined PfMORC's interactions with proteins, its genomic distribution in different parasite life stages (rings, trophozoites, schizonts), and the transcriptome's response to PfMORC depletion. They conducted a chromatin conformation capture on PfMORC-depleted parasites and observed significant alterations. Furthermore, they demonstrated that PfMORC depletion is lethal to the parasite.Strengths: This study significantly advances our understanding of PfMORC's role in establishing heterochromatin. The direct consequences of the PfMORC depletion are addressed using chromatin conformation capture.

We appreciate the Reviewer’s comments and reflection on the importance of our work.

Weaknesses: The study only partially addressed the direct effects of PfMORC depletion on other heterochromatin markers.

Here again, we agree with the reviewer’s comment and have performed additional experiments to delve deeper into the multifaceted roles of _Pf_MORC. We have performed additional ChIP-sequencing analysis on _Pf_MORC depleted conditions focusing on known heterochromatin and euchromatin markers H3K9me3 and H3K9ac respectively. We hope our new results presented in figure 5 will shed light on the more direct implications of _Pf_MORC on heterochromatin and gene silencing.

**Reviewer #1 (Recommendations For The Authors):**
Suggestions for improved or additional experiments, data or analyses.• Why does MORC, which was used in the pull-down, seem to be only minimally enriched in the volcano plot, while a series of proteins (marked in red) and AP2 (highlighted in green) are enriched with log2 fold changes exceeding 15?

We apologize for the confusion. MORC was detected with the highest number of peptides (97 and 113) and spectra (1041 and 1177) confirming the efficiency of our pull-down. However, considering the relatively large size of the MORC protein (295kDa) and it weak detection in the control (5 and 7 peptides; 16 and 43 spectra), the Log2 FoldChange and Z-statistic after normalization are minimal compared to smaller proteins that were not identified in the control samples.

Additionally, can you explain why these proteins appear to be enriched at the same fold?

We can postulate that these proteins form a complex with a ratio of 1:1. Two of these three proteins are described to interact with MORC in several publications, supporting a strong interaction between them.

Variations in the interactome could result from the washing buffer's stringency.

We agree that the IP conditions could affect the detection of the interactome as well as the parasite stage used. As indicated below, the overlap with previous publications and the presence of AP2 TFs and chromatin remodelers strongly support our results.

It would be highly appropriate for the authors, similar to the co-submitted article (Maneesh Kumar Singh et al.), to present their mass spectrometry data in relation to previous purifications in Plasmodium (Bryant et al. 2020; Subudhi et al. 2023; Hillier et al. 2019) and also in Toxoplasma (Farhat et al. 2020). It would be good if authors could also put their results into perspective in light of the following pre-prints:

We agree with the reviewer’s comment. In this revised manuscript, we compared our IP-MS data to previous published manuscripts. Key proteins including the AP2-P (PF3D7_1107800) and HDAC1 were indeed identified in several experiments validating our initial findings of the formation of large complexes with MORC. However, it’s important to highlight that the MORC protein was not used as the bait protein in previously published papers, and thus some discrepancies can be observed.

Given the tendency of MORCs to form multiple complexes with AP2 factors, have you explored whether specific AP2s are conserved between Plasmodium and Toxoplasma, within the phylum?

*P. falciparum* encodes for 27 putative AP2s, while *T. gondii* has over 60 AP2s, making direct comparison challenging. Some *Plasmodium* AP2s have multiple counterparts in *T. gondii* and typically conservation is limited to the AP2 binding domains. Attempts to identify sequence homology among AP2s and the regions of conservation have been performed (PMID: 30959972, PMID: 30959972, PMID: 16040597). Although this information would provide interesting insight, we believe exploring this topic at this time would diverge from our primary objectives. It would be more appropriate to address this in future studies.

Could this conservation be identified either through phylogenetic means or by using tools such as AlphaFold, especially considering not just the AP2 domains but also any existing ACDC domains?

Although this may reveal important information regarding the association between MORC proteins and AP2 domains, we believe investigating the conservation between AP2 across apicomplexan parasites may prove too challenging and is beyond the scope of this work.

Most of the genes are depicted without their immediate surroundings (Fig. 2d and Fig S2c, d). For instance, the promoter region of AP2g is not shown (Fig. 2d). It is therefore very challenging to determine the presence or absence of MORC upstream or downstream; considering that this factor, which can create DNA loop protrusions, might bind at a distance from the genes in question.

All gene coverage plots, including AP2-G, show 500 bp up- and downstream of the displayed gene. We have modified our figure legends to make sure that this information is provided.

Upon examining Figure S3, it is evident that the authors have indicated a decline in PfMORC expression, represented as percentages over two unique time frames. The methodology behind this quantification remains ambiguous. It's essential for the authors to specify whether normalization was done using a loading control. As a benchmark, Singh et al. (2021) in their Figure 4 transparently used GAPDH as a loading control and included an untreated sample in their western blot analysis.

We thank the Reviewer for bringing this to our attention. Our initial quantification was performed using ImageJ. To address the Reviewer’s comment, we have reperformed the experiment. Our quantitative analysis was performed through Bio-Rad ImageLab software using aldolase expression as a loading control (50% of the MORC loading). This information has now been incorporated into the supplementary figures (Figure S3).

There's a striking observation that, despite significant degradation of PfMORC (as depicted in Figures S1 and S3), only the upper band in the western blot diminishes. This inconsistency needs addressing, as it can raise questions about the interpretation of the results.

We agree with the reviewer's comment. We experienced some challenges upon performing a Western Blot on such a large protein (295kDa). Our initial attempts required long exposure that may have highlighted non-specific signals of smaller proteins. To address the reviewer’s comment, we have performed the experiment one more time and made necessary changes to our WB protocol. Our new result better reflects the expected down regulation of _Pf_MORC. These changes have been incorporated to our manuscript and Fig S3.

Recommendations for improving the writing and presentation.MORC KD quantification and consistency with previous findings (Figure S3): When comparing their results with those from another study (Singh et al. 2021), it's critical to ensure that the experimental conditions, especially the methodology for KD and the quantification of protein levels, are similar. If not, a direct comparison might be misleading.

We greatly appreciate the suggestions and have made efforts to redesign the MORC KD quantifications according to the reviewer’s recommendations.

While the manuscript mentions the level of KD, it does not delve into the functional consequences of such a decrease in protein levels. It would be of interest to understand how this level of KD affects the parasite's biology, especially in the context of the paper's main findings.

We have addressed this question by looking at the changes in chromatin structure in WT versus KD parasites upon atc removal. We have also validated this initial result by designing an additional ChIP-seq experiment against histone marks in WT versus KD parasites upon atc removal. Our findings showed a significant downregulation in H3K9me coverage in heterochromatin regions, specifically in genes associated with antigenic variation and invasion genes. These findings suggest that PfMORC regulates at least partially gene silencing and chromatin arrangements. The manuscript has been edited accordingly.

Concluding page 5, the authors present an interpretation of their findings that suggests a multi-faceted role of PfMORC in regulating stage-specific gene families, particularly the gametocyte-related genes and merozoite surface proteins. While the narrative they present is intriguing, several concerns arise:Over-reliance on correlation: The authors draw a direct line between the levels of PfMORC binding and the function of these genes in the parasite's life cycle. However, a mere correlation between PfMORC binding and stage-specific gene activity does not necessarily imply causation. They would need to provide experimental evidence showing that manipulation of PfMORC levels directly impacts these genes' expression.

We agree with the reviewer's comment. We have however partially addressed this issue by comparing our ChIP-seq, RNA-seq and Hi-C experiments. We concluded that several of the transcriptional changes observed were due to an indirect effect of PfMORC KD and were most likely induced by a cell cycle arrest and partial collapse of the chromatin structure. The collapse of the heterochromatin structure was validated using our Hi-C experiment. To further address additional concerns the review’s had, we have included additional ChIP-seq experiments targeting histone marks to confirm our initial hypothesis. Result of this additional experiment has been incorporated in the revised version of the manuscript.

Ambiguity surrounding "low levels" and "high levels": The terms "low levels" and "high levels" of PfMORC binding are qualitative and could be subject to interpretation. Without quantification or a clear benchmark, these descriptions remain vague.

We agree with the reviewers that the terms "low levels" and "high levels" of PfMORC binding are qualitative and could be subject to interpretation. We have however quantified our change in DNA binding using normalized reads (RPKM). In trophozoite and schizont stages, most of the genes contain a mean of <0.5 RPKM normalized reads per nucleotide of _Pf_MORC binding within their promoter region, whereas antigenic gene families such as *var* and *rifin* contain ~1.5 and 0.5 normalized reads, respectively (Fig. 2b). Similar results are also obtained for the gametocyte-specific transcription factor AP2-G that contains levels of _Pf_MORC binding similar to what is observed in *var* genes (Fig. 2c and S2c, d).

Shift in Binding Sites: The observed minor switch in PfMORC binding sites from gene bodies to intergenic and promoter regions is mentioned, but without context on how these shifts impact gene expression or any comparative analysis with other proteins showing similar shifts. The claim that this shift implicates PfMORC as an "insulator" is a leap without direct evidence.

We apologize for the confusion. We have compared our ChIP-seq with RNA seq results at different time points of the cell cycle and demonstrated that the shift observed has an effect in gene expression. We have edit the manuscript to clarify these results.

Overextension of PfMORC's Role: The authors suggest that PfMORC moves to the regulatory regions around the TSS to guide RNA Polymerase and transcription factors. This is a substantial claim and would require additional experiments to validate. Simply observing binding in a region is insufficient to assign a specific functional role, especially one as critical as guiding RNA Polymerase. Historically, the MORC family has been primarily linked with gene silencing across Apicomplexan, plants, and metazoans. On page 7, the authors noted a minimal overlap between the ChIP-seq and RNA-seq signals (Fig. 4e). They also acknowledged that the pronounced gene expression shifts at schizont stages result from a combination of direct and indirect impacts of PfMORC degradation, which could cause cell cycle arrest and potential heterochromatin disintegration, rather than just decreased PfMORC binding. Therefore, the authors should adjust their conclusions in the manuscript to more accurately represent the multifaceted functions of MORC in the parasite.

We agree with the reviewer's comment and have edited the manuscript accordingly.

DISCUSSION:The authors concluded that "Using a combination of ChIP-seq, protein knock down, RNA-seq and Hi-C experiments, we have demonstrated that the MORC protein is essential for the tight regulation of gene expression through chromatin compaction, preventing access to gene promoters from TFs and the general transcriptional machinery in a stage specific manner."Again, the assertion that MORC protein is essential for tight regulation of gene expression, based purely on correlational data (e.g., ChIP-seq showing binding doesn't prove functionality), assumes causality which might not be fully substantiated. The phrase "preventing access to gene promoters from TFs and the general transcriptional machinery in a stage-specific manner" needs also validation. Asserting that MORC is essential for this function might oversimplify the process and overlook other critical contributors.

We agree with the reviewer’s comments and the conclusion has since been edited accordingly.

The discussion is quite poor. It would be pertinent to put MORC in perspective within the broader picture of regulatory mechanisms of chromatin state at telomeres and var genes. For instance, how do SIR2 and HDAC1 (associated with MORC) divide the task of deacetylation? Or the contribution of HP1 and other non-coding RNAs.

We agree with the reviewer’s suggestion. However, in order to put MORC in perspective within a broader picture, we would need to measure changes in localization of several molecular components regulating heterochromatin in WT versus KD condition. This will require access to several molecular tools and specific antibodies that we do not currently have. We have addressed these issues in our discussion.

Minor corrections to the text and figures.Figure 1d: Could you provide the ID for each AP2 directly on the volcano plot? While some IDs are referenced in the manuscript, visual representation in the plot would facilitate a clearer understanding of their enrichment levels.

ID for unknown AP2 proteins have been added on the volcano plot.

I recommend presenting Figure S2b as a panel within a primary figure. This change would offer readers a more quantitative understanding of the distinct differences between developmental stages. Notably, there seems to be a limited number of genes in common when considering the total, and there is an apparent lack of enrichment in the ring stage.

This has been done.

The captions are very minimally detailed. An effort must be made to better describe the panels as well as which statistical tests were used.

We have improved the figure legends and add the number of biological replicates as well as the statistic used in each figure legend.

Figure 1A: The protein diagram with its domains does not take scale into account.

The figure has been modified.

**Reviewer #2 (Recommendations For The Authors):**
(1) The study lacks a direct link between PfMORC's inferred function and the state of heterochromatin in the genome post-depletion.

We agree with the reviewer's comment and have included additional ChIP-seq experiments to measure changes in histone marks in PfMORC depleted parasite line. We show a significant decrease in histone H3K9me3 marks in PfMORC KD condition.

Conducting ChIP-seq on well-known heterochromatin markers such as H3K9me3, HP1, or H3K36me2/3 could shed light on the consequences of PfMORC depletion on global heterochromatin and its boundaries.

With no access to an anti-HP1 antibody with reasonable affinity, we have not been able to study the impact of MORC KD on HP1 but have successfully observed the impact on H3K9me3 marks. These results have been added to the revised manuscript in (Fig. 5).

(2) The authors should conduct a more comprehensive analysis of PfMORC's genomic localization, comparing it to ApiAP2 binding (interacting proteins) and histone modifications. This would provide valuable insights.

We have performed a more comprehensive genome wide analysis of MORC binding through ChIP-seq on WT and MORC-KD conditions. Our results show that _Pf_MORC localizes to heterochromatin with significant overlap with H3K9-trimethylation (H3K9me3) marks, at or near *var* gene regions. When downregulated, level of H3K9me3 was detected at a lower level, validating a possible role of _Pf_MORC in gene repression. Regarding the comparison with AP2 binding, our proteomics datasets have shown extensive MORC binding with several AP2 proteins.

(3) RNA-seq data reveals that only a few genes are affected after 24 hours of PfMORC depletion, with an equivalent number of up-regulated and down-regulated genes. The reasons behind down-regulation resulting from a heterochromatin marker depletion are not clearly established.

We agree with the reviewer’s comment. At this stage (24 hours), _Pf_MORC depletion is limited and the effects at the transcriptional level are quite restricted. Furthermore, it is highly probable that down-regulated genes are most likely due to an indirect effect of a cell cycle arrest. We have edited the manuscript to address this comment.

The relationship between this data and the partial depletion of PfMORC needs further discussion.

We agree with the reviewers and have improved our discussion in the revised version of the manuscript.

(4) The authors did not compare their ChIP-seq data with the genes found downregulated in the RNA-seq data. Examining the correlation between these datasets would enhance the study.

We apologize for the confusion. We have compared ChIP-seq and RNA-seq data and identified a very limited number of overlapping genes indicating that most of the changes observed in gene expression are in fact most likely indirect due to a cell cycle arrest and a collapse of the chromatin. We have edited the manuscript to clarify this issue.

(5) The discussion section is relatively concise and does not fully address the complexity of the data, warranting further exploration.

We have improved the discussion section in the revised version of the manuscript.